Corrected: Author correction

# Crizotinib-induced immunogenic cell death in non-small cell lung cancer

Peng Liu [1,2,3,4,5,6], Liwei Zhao[1,2,3,4,5,6], Jonathan Pol[2,3,4,5,6], Sarah Levesque[1,2,3,4,5,6], Adriana Petrazzuolo[1,2,3,4,5,6], Christina Pfirschke[7], Camilla Engblom[7,8], Steffen Rickelt [9], Takahiro Yamazaki[10], Kristina Iribarren[2,3,4,5,6], Laura Senovilla[2,3,4,5,6], Lucillia Bezu[1,2,3,4,5,6,11], Erika Vacchelli[2,3,4,5,6], Valentina Sica[2,3,4,5,6], Andréa Melis[12,13], Tiffany Martin[12,13], Lin Xia[14,15], Heng Yang[14,15], Qingqing Li[14,15], Jinfeng Chen[14,15], Sylvère Durand[2,3,4,5,6], Fanny Aprahamian[2,3,4,5,6], Deborah Lefevre[2,3,4,5,6], Sophie Broutin[16], Angelo Paci[16,17], Amaury Bongers[16], Veronique Minard-Colin[18], Eric Tartour [19,20], Laurence Zitvogel[1,15,18,21,22], Lionel Apetoh[12,13], Yuting Ma [14,15], Mikael J. Pittet[7,23], Oliver Kepp[1,2,3,4,5,6] & Guido Kroemer[2,3,4,5,6,15,24,25]

Immunogenic cell death (ICD) converts dying cancer cells into a therapeutic vaccine and stimulates antitumor immune responses. Here we unravel the results of an unbiased screen identifying high-dose (10 µM) crizotinib as an ICD-inducing tyrosine kinase inhibitor that has exceptional antineoplastic activity when combined with non-ICD inducing chemotherapeutics like cisplatin. The combination of cisplatin and high-dose crizotinib induces ICD in non-small cell lung carcinoma (NSCLC) cells and effectively controls the growth of distinct (transplantable, carcinogen- or oncogene induced) orthotopic NSCLC models. These anticancer effects are linked to increased T lymphocyte infiltration and are abolished by T cell depletion or interferon-γ neutralization. Crizotinib plus cisplatin leads to an increase in the expression of PD-1 and PD-L1 in tumors, coupled to a strong sensitization of NSCLC to immunotherapy with PD-1 antibodies. Hence, a sequential combination treatment consisting in conventional chemotherapy together with crizotinib, followed by immune checkpoint blockade may be active against NSCLC.

[1] Faculty of Medicine, University of Paris Sud, Kremlin-Bicêtre 94270, France. [2] Cell Biology and Metabolomics Platforms, Gustave Roussy Cancer Campus, Villejuif 94805, France. [3] Equipe 11 labellisée Ligue Nationale contre le Cancer, Centre de Recherche des Cordeliers, Paris 75006, France. [4] Institut National de la Santé et de la Recherche Médicale, UMR1138, Equipe labellisée Ligue Nationale Contre le Cancer, Paris 75006, France. [5] Université Paris Descartes, Sorbonne Paris Cité, Paris 75006, France. [6] Université Pierre et Marie Curie, Paris 75006, France. [7] Center for Systems Biology, Massachusetts General Hospital Research Institute and Harvard Medical School, Boston 02139 MA, USA. [8] Graduate Program in Immunology, Harvard Medical School, Boston 02238 MA, USA. [9] Koch Institute for Integrative Cancer Research, Massachusetts Institute of Technology, Cambridge 02139 MA, USA. [10] Department of Radiation Oncology, Weill Cornell Medical College, New York 14853 NY, USA. [11] Department of Anaesthesiology, Hôpital Européen Georges Pompidou, Paris 75015, France. [12] Centre de Recherche INSERM LNC-, UMR1231 Dijon, France. [13] Department of Medical Oncology, Centre Georges-François Leclerc, Dijon 21000, France. [14] Center for Systems Medicine, Institute of Basic Medical Sciences, Chinese Academy of Medical Sciences and Peking Union Medical College, Beijing 100730, China. [15] Suzhou Institute of Systems Medicine, Suzhou 215123 Jiangsu, China. [16] Department of Pharmacology, Institut Gustave Roussy, Villejuif 94805, France. [17] School of Pharmacy, Université Paris Sud, Châtenay-Malabry 92 296, France. [18] Institut de Cancérologie, Gustave Roussy Cancer Campus (GRCC), Villejuif 94805, France. [19] INSERM U970, Université Paris Descartes Sorbonne Paris-Cité, Paris 75006, France. [20] Department of Immunology, Hôpital Européen Georges Pompidou, Paris 75015, France. [21] INSERM U1015, Villejuif 94805, France. [22] Center of Clinical Investigations CIC1428, Villejuif 94805, France. [23] Department of Radiology, Massachusetts General Hospital, Boston 02114 MA, USA. [24] Pôle de Biologie, Hôpital Européen Georges Pompidou, AP-HP, Paris 75015, France. [25] Department of Women's and Children's Health, Karolinska University Hospital, Stockholm 141 86, Sweden. These authors contributed equally: Peng Liu, Liwei Zhao. These authors jointly supervised this work: Oliver Kepp, Guido Kroemer. Correspondence and requests for materials should be addressed to O.K. (email: captain.olsen@gmail.com) or to G.K. (email: kroemer@orange.fr)

Several druggable oncogenes are tyrosine kinases that become activated due to mutations or gene amplifications. In consequence, small inhibitory molecules and antibodies targeting such tyrosine kinases (when they are expressed on the cell surface) have been developed and introduced into the clinics. Indeed, tyrosine kinase inhibitors (TKI) targeting oncogenes such as *cABL* (activated in Philadelphia chromosome-positive chronic myeloid leukemia, CML)[1], *BRAF* (activated in melanoma)[2], ERBB2 (activated in a fraction of breast cancers)[3], *EGFR* (activated in a sizable portion of non-small cell lung cancers, NSCLC)[4], *KIT* (activated in gastrointestinal stromal tumors, GIST)[5], or *VGFR* (activated in renal cancers and others)[6], have been approved for the routine treatment of cancer patients.

The development of anti-neoplastic TKIs has been largely driven by the cell-autonomous view that (i) cancer is a genetic and epigenetic cellular disease and (ii) anticancer drugs should target specific characteristics of transformed cells to eliminate them or to reduce their growth[7]. At odds with this vision, however, imatinib mesylate, the first TKI to be introduced into routine praxis, initially for the treatment of CML (if positive for the *ABL* activating translocation or activating mutations of *KIT*)[4] and later for that of GIST expressing activated *KIT*[5], turned out to mediate (some of) its effects via the stimulation of NK and T cell-mediated anticancer immune responses[8]. Indeed, in the treatment of GIST, signs of NK cell activation and the isoform expression pattern of NK cell-activating receptors constitute accurate biomarkers that predict the long-term outcome of the treatment beyond its discontinuation[8,9]. Thus, imatinib exemplifies a TKI that, in addition to its direct effects on malignant cells, stimulates the cellular immune system, presumably through the inhibition of unmutated KIT expressed by dendritic cells (DC)[10]. It is indeed a subject of debate to which extent the success of imatinib and that of second-generation TKIs targeting ABL and KIT may be attributed to direct and immune-dependent effects[8]. Importantly, other TKIs have also been shown to stimulate anticancer immune responses, as it applies to sorafinib[11] and ibrutinib[12], among others[7].

Chemotherapeutic agents like anthracyclines, oxaliplatin, taxanes, and cyclophosphamide are also known to mediate part of their anticancer effects through indirect, immune-dependent mechanisms[7,13–16]. These cytotoxicants can cause cell death that is preceded by *premortem* stress responses, allowing the cancer cells to emit signals that render them detectable for the immune system[17]. This 'immunogenic cell death' (ICD) is characterized by an autophagic response that allows the cells to release ATP during the blebbing phase of apoptosis or during necrotic demise[15], as well as an endoplasmic reticulum (ER) stress response (with phosphorylation of eIF2α as a prominent hallmark) causing exposure of calreticulin (CALR) on the cell surface[17]. ATP acts as a chemoattractant for DC precursors expressing purinergic receptors[18], while CALR functions as an 'eat me' signal to facilitate the phagocytosis of portions of the dying cancer cell (with the tumor-associated antigen) by the DC[19]. Cell death is also associated with the release of the cytoplasmic protein annexin A1 (ANXA1, which acts as a chemotactic factor on formyl peptide receptor-1, FPR1, for assuring DC to form synapses with dying cells)[20] and the nuclear protein high mobility group box 1 (HMGB1, which serves as a DC maturation factor by activating Toll-like receptor-4, TLR4)[21]. Clinical evidence has been obtained in favor of the importance of ICD and of each of the aforementioned ligands and receptors, meaning that malignant cells lacking features of ICD (such as autophagy, CALR, and HMGB1) or hosts with deficient FPR1 or TLR4 have reduced chances of progression-free or overall survival post-chemotherapy[17]. There is also evidence that cisplatin (CDDP), mitomycin C (MitoC) or other prominent chemotherapeutics are relatively inefficient due to their incapacity to stimulate ICD[7,17]. Thus, measures to improve ICD induction can improve the efficacy of CDDP and MitoC in preclinical models, as well as in patients[22].

Recent evidence pleads in favor of the idea that several therapeutic antibodies targeting surface-expressed TKIs also induce ICD, suggesting that their clinical efficacy is dictated by immune mechanism as well[23,24]. However, thus far no small molecule TKI have been shown to induce ICD. Based on this consideration, we developed a screen to identify TKIs that might stimulate the hallmarks of ICD (such as autophagy, CALR exposure, and HMGB1 exodus). Here we show that crizotinib, an agent that is used to treat NSCLC carrying activated ALK and ROS1, acts as a potent ICD stimulator through off-target effects. We provide preclinical evidence that crizotinib can be advantageously combined with non-ICD inducing chemotherapeutics, as well as with immune checkpoint blockade, to treat NSCLC that lack genetic rearrangements leading to the activation of ALK or ROS1.

## Results

**Identification of (R)-crizotinib as a potential ICD inducer**. We used a fully automatized high-content, medium throughput fluorescence-based screening procedure[22] to identify potential ICD inducers among two libraries of tyrosine kinase inhibitors (TKIs), namely the *Public Chemogenomic Set for Protein Kinases* with more than 500 compounds[25] (Fig. 1a) and an in-house collection of FDA-approved TKIs (32 compounds) that are currently used in the oncological armamentarium (Fig. 1b). Human osteosarcoma U2OS cells expressing CALR-red fluorescent protein (RFP), green fluorescent protein (GFP)-LC3 or HMGB1-GFP fusion proteins were treated with each of these compounds at a concentration that is high enough to induce off-target effects (10 μM). Robotized fluorescence microscopy and automated high-content image analysis were used to determine surrogate markers of ICD such as the redistribution of CALR-RFP into peripheral dots, the generation of autophagy-associated GFP-LC3 *puncta* in the cytoplasm and the exodus of HMGB1-GFP from the nucleus into the cytosol, as well pyknosis of nuclei counterstained with DAPI. Of note, both screens yielded an overlapping set of TKIs that were able to induce ICD characteristics to a similar level as the control, the anthracycline mitoxantrone (MTX), namely, (R)-crizotinib, foretinib, canertinib, lestaurtinib and ceritinib (Fig. 1a, b). This panel of TKIs indeed induced signs of ICD such as immunofluorescence-detectable CALR exposure and the release of ATP and HMGB1 into the culture supernatant in several human cancer cell lines (U2OS, cervical carcinoma HeLa, colorectal cancer HCT-116) and in murine fibrosarcoma MCA205 cells (Supplementary Fig. 1a–c).

The aggregate analysis of the screening experiments (Fig. 1) ranked (R)-crizotinib first among FDA-approved TKIs. Therefore, we decided to concentrate our efforts on (R)-crizotinib. Of note, (R)-crizotinib was more efficient than its enantiomer (S)-crizotinib in inducing CALR-RFP redistribution and pyknosis in U2OS cells (Fig. 1c, d), as well as in mouse fibrosarcoma MCA205 and lung adenocarcinoma TC1 cells (Fig. 1d). When combined with poor ICD inducers such as cisplatin (CDDP) and mitomycin C (MitoC), both of which lack the ability to cause CALR exposure[13,14], (R)-crizotinib (but not (S)-crizotinib) was able to reestablish CALR exposure to the same level as that observed for MTX. This result was obtained in several cancer cell lines including U2OS, MCA205 and TC1 cells (Fig. 1d). (R)-crizotinib and several other clinically used ALK inhibitors (such a certinib, entrectinib and others) induced the phosphorylation of eIF2α (Supplementary Fig. 1d) and ICD hallmarks in U2OS cells (Supplementary Fig. 1e–g).

As expected, (R)-crizotinib (but not (S)-crizotinib) was more efficient in inducing apoptosis and ICD hallmarks (CALR exposure, ATP and HMGB1 release) when added to a human

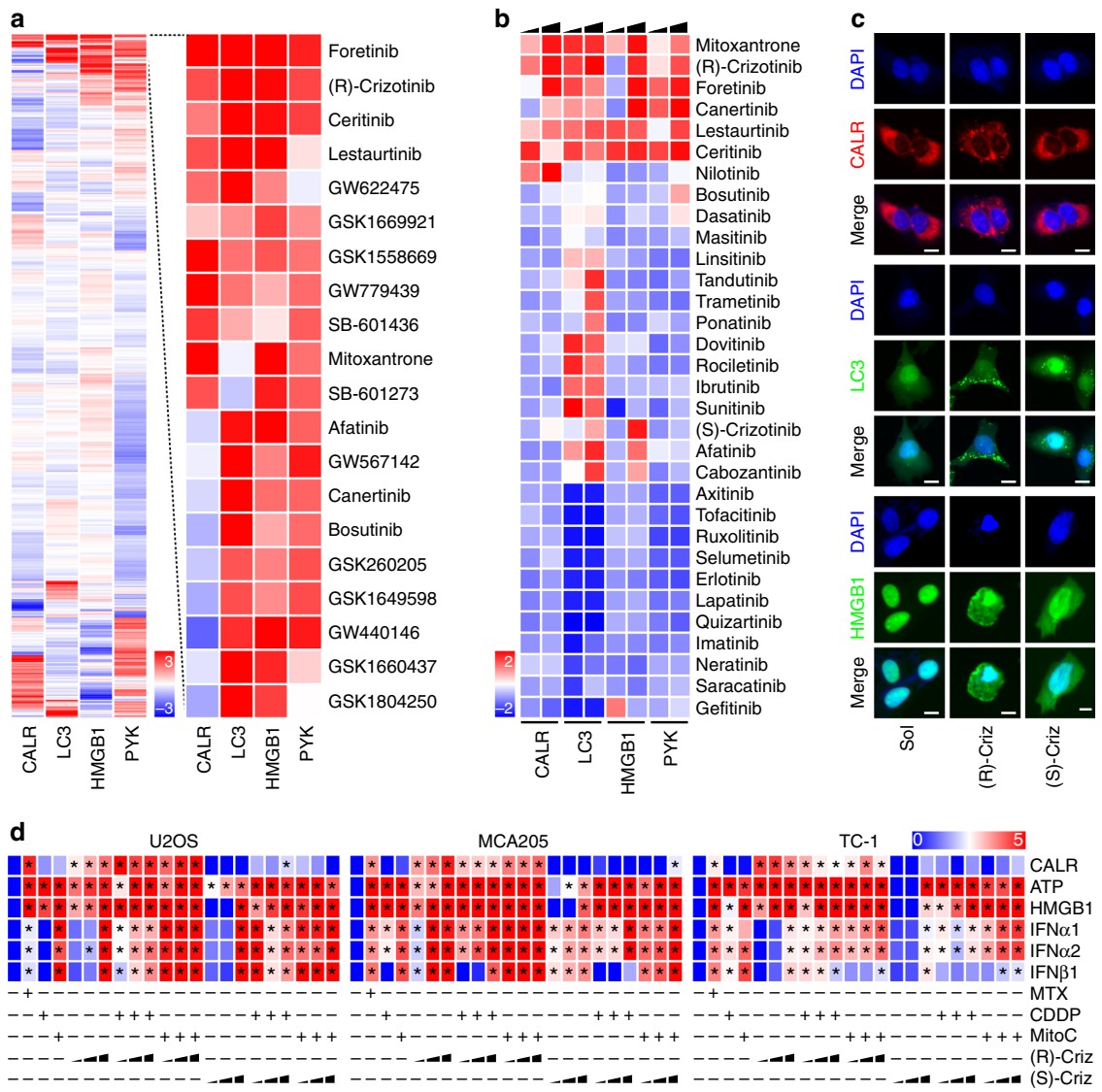

**Fig. 1** Identification of (R)-crizotinib as a novel immunogenic cell death inducer. **a** Human osteosarcoma U2OS cells stably co-expressing calreticulin (CALR)-RFP and HMGB1-GFP or GFP-LC3 were treated with the compounds from the *Public Chemogenomic Set for Protein Kinases* at a concentration of 10 μM for 8 h, 12 h and 32 h, followed by the assessment of CALR exposure (CALR-RFP redistribution), autophagy (GFP-LC3 granularity) and HMGB1 release (decrease of nuclear HMGB1-GFP intensity), respectively. Nuclear pyknosis was monitored as an indicator for cell death. The data was normalized to calculate Z scoring (mean, $n = 4$, Supplementary data 1) and the effects of the agents are hierarchically clustered in a heatmap. Top hits are separately displayed with red and blue reporting positive and negative values, respectively. **b** A collection of tyrosine kinase inhibitors (final concentrations 5, 10 μM) was screened following the same approach. Z score was calculated for each agent (mean, $n = 4$, Supplementary data 2). **c** Representative images of (R)-crizotinib (Criz) induced ICD phenotypes are depicted and the scale bar equals 10 μm. **d** ICD inducing effects of (R)/(S)-crizotinib in combination with cisplatin (CDDP; 150 μM) and mitomycin C (MitoC; 150 μM). Human osteosarcoma U2OS cells, murine fibrosarcoma MCA205 cells, as well as murine NSCLC TC1 cells were treated with mitoxantrone (MTX; 2 μM), CDDP, MitoC alone or in combination with increasing concentrations (1, 5, 10 μM) of (R) or (S)-crizotinib, for 24 h before determination CALR exposure (by flow cytometry), ATP secretion (ATP-dependent luminescence kit), and HMGB1 release (ELISA). Results are normalized as log2 and shown as a heatmap in which each rectangle represents the mean of triplicate assessment, statistical significances are indicated as $*p < 0.001$ comparing to controls using Student's *t*-test

NSCLC cell line that is positive for the EML4-ALK fusion protein (H2228 cells) than when administered to cells that are negative for the ALK activating translocation (H1650 cells) (Fig. 2a–h). However, at high doses of 5–10 μM (R)-crizotinib was active on cells lacking the EML4-ALK fusion protein (Fig. 2a, c, e, g). This contrasts with the fact that metabolic effects such as glycolysis and mitochondrial respiratory inhibition by (R)-crizotinib (Supplementary Fig. 2a–f), or downregulation of hexokinase-2 were (Supplementary Fig. 2g, h) exclusively detectable in EML4-ALK-positive H2228 cells[26]. (R)-crizotinib induced eIF2α phosphorylation, a sign of endoplasmic reticulum stress associated

with CALR exposure[27], in U2OS cells (Supplementary Fig. 3a) and in both H1650 and H2228 cells, though with distinct kinetics (Supplementary Fig. 3b, c).

In U2OS cells, the depletion of several (R)-crizotinib-inhibited tyrosine kinases, namely ALK, JAK2, MET and ROS1 by individual small interfering RNAs (siRNAs) or pools of such siRNAs (validated by RT-PCR. Supplementary Fig. 3d) triggered several hallmarks of ICD to the same level as did (R)-crizotinib (Fig. 2i–l). Thus, depletion of ALK, JAK2, MET, and ROS1 induced significant eIF2α phosphorylation (Fig. 2i), redistribution of CALR-RFP to dots (Fig. 2k) and HMGB1-GFP release into the

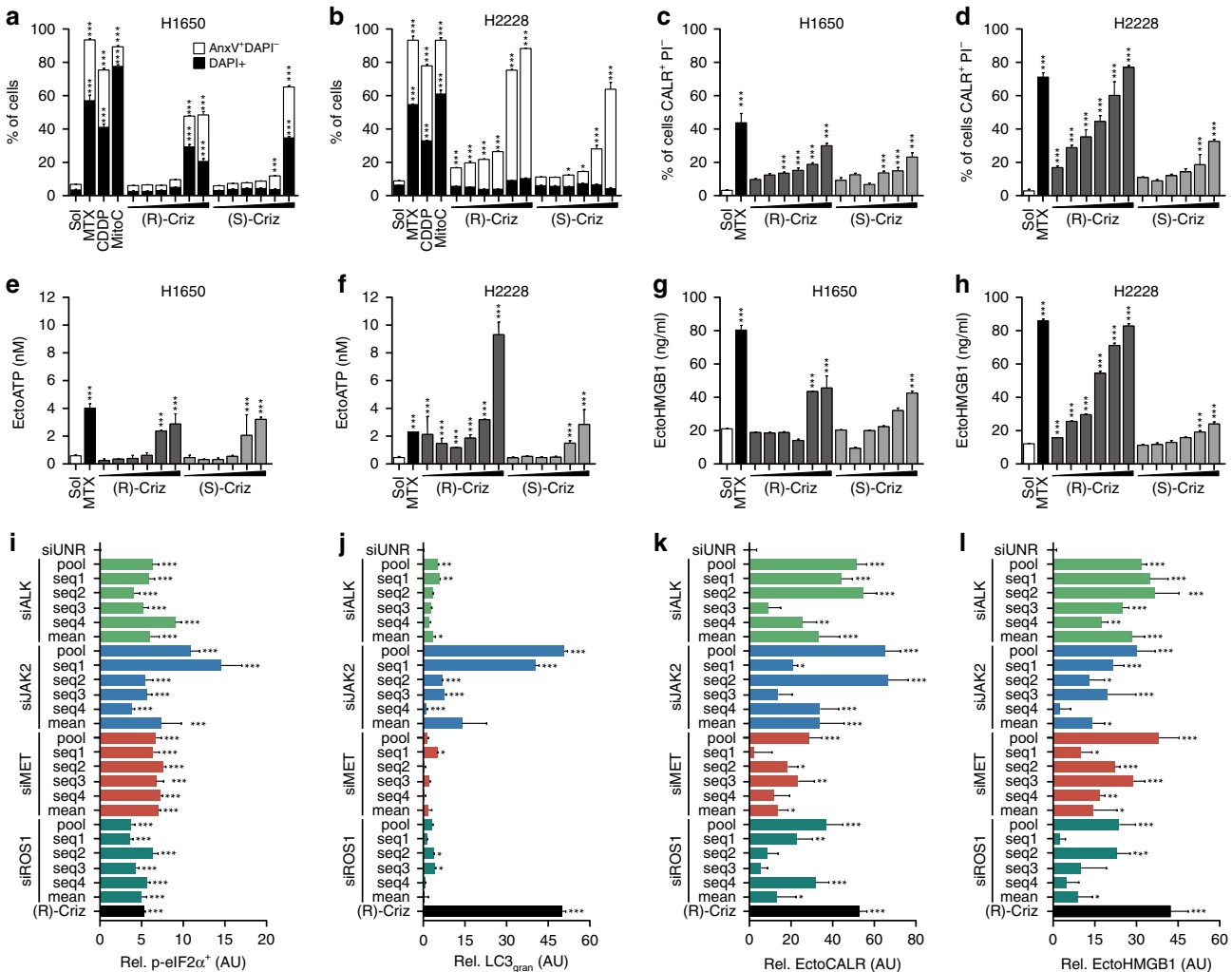

**Fig. 2** Mode of action of (R)-crizotinib induced ICD. **a–h** Investigation of crizotinib induced ICD in the NSCLC cell lines, NCI-H2228 and NCI-H1650, which are positive and negative for the EML4-ALK fusion protein, respectively. H1650 (EML4-ALK⁻) and H2228 (EML4-ALK⁺) cells were treated with mitoxantrone (MTX; 2 µM), or increasing concentrations (0.1, 0.5, 1, 2.5, 5, 10 µM) of (R)-crizotinib or (S)-crizotinib (Criz) for 72 h to assess cell death using AnnexinV, DAPI staining (**a**, **b**), and 24 h to determine immunogenic cell death parameters (**c–h**). **i–l** SiRNAs knockdown of crizotinib targeting kinases induces ICD similar to crizotinib. Indicated siRNAs (a pool of four siRNA duplexes or 4 individual sequences) were reversely transfected into U2OS cells or their derivatives stably co-expressing calreticulin (CALR)-RFP and HMGB1-GFP or GFP-LC3. Forty-eight h later the phosphorylation of eIF2α and other ICD hallmarks (CALR-RFP redistribution, GFP-LC3 granularity and HMGB1-GFP intensity) were assessed. Treatment of unrelated-siRNA-transfected cells with (R)-crizotinib for 24 h was used as a positive control. The capacity of siRNA-mediated knockdowns to induce ICD parameters was normalized to unrelated-siRNA-transfected cells. Data are shown as mean ± s.e.m. ($n = 3$; *$p < 0.05$, **$p < 0.01$, ***$p < 0.001$, compared to unrelated siRNA using Student's $t$-test)

cytoplasm (Fig. 2l). In contrast, only the depletion of JAK2 was as efficient as (R)-crizotinib in inducing the formation of GFP-LC3 *punctae* (Fig. 2j). We knocked down several other genes coding for tyrosine kinases that are not inhibited by (R)-crizotinib (such as BTK, EGFR, ERBB, HCK) and found that these manipulations induced less CALR exposure, ATP release and HMGB release than treatment with (R)-crizotinib (Supplementary Fig. 3e–g). Thus, it appears that the inhibition of a specific set of TKIs by (R)-crizotinib participates in ICD induction. Confirming a prior report[28], (R)-crizotinib also induced a higher expression of class I and class II MHC molecules (Supplementary Fig. 3h–k). In contrast, (R)-crizotinib failed to significantly modulate mRNA expression of *ALK*, *JAK2*, *MET*, and *ROS1* (Supplementary Fig. 3l–n).

**(R)-crizotinib plus cisplatin induces ICD in vivo.** As previously reported[13,14,20], in vitro MTX-treated TC1 cells can be injected

subcutaneously (s.c.) to protect immunocompetent C57BL/6 mice against rechallenge with live tumor cells of the same kind injected two weeks later into the opposite flank (Fig. 3a, b). This contrasts with CDDP (Fig. 3c, e) or MitoC (Fig. 3d, e)-treated TC1 cells that largely failed to induce ICD in this in vivo assay. (R)-crizotinib-treated TC1 cells formed tumors at the injection site (not shown), correlating with the fact that, in contrast with chemotherapeutic agents such as CDDP, (R)-crizotinib alone was unable to completely abolish the clonogenic potential of tumor cells (Supplementary Fig. 4). However, TC1 cells became immunogenic when they were cultured in vitro with either CDDP combined with (R)-crizotinib (Fig. 3c, e) or MitoC together with (R)-crizotinib (Fig. 3d, e). The gene editing-mediated invalidation of the *Anxa1* or *Hmgb1* genes in TC1 cells was compatible with the survival of these cells, as well as the formation of tumors in vivo. However, in contrast to WT cells, *Anxa1*⁻/⁻ or *Hmgb1*⁻/⁻ cells treated with CDDP plus (R)-crizotinib in vitro failed to elicit anticancer immune responses against TC1 cells (Fig. 3f, g). Similarly, CALR

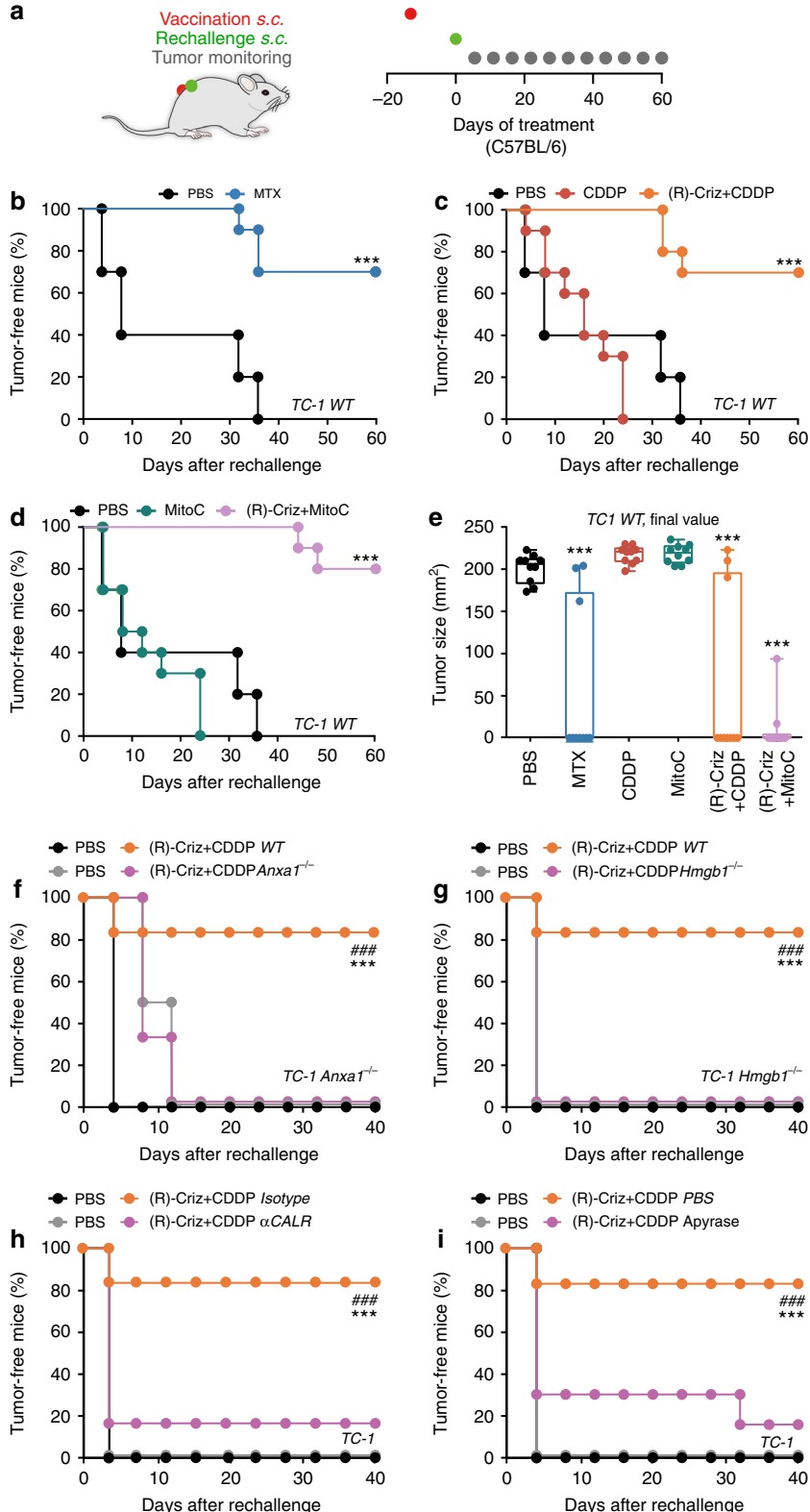

neutralization with a blocking antibody[14] or the destruction of extracellular ATP with apyrase[29], abolished the immunogenicity of TC1 cells killed by CDDP plus (R)-crizotinib (Fig. 3h, i).

Very similar results were obtained when this in vitro/in vivo protocol for assessing ICD was applied to MCA205 cells. Again, (R)-crizotinib was highly efficient in inducing ICD when combined with CDDP or MitoC (Supplementary Fig. 5a–d) and knockout of Anxa1 or Hmgb1, blockade of CALR or hydrolysis of

ATP abolished the immunogenicity of MCA205 cells (Supplementary Fig. 5e–h). Direct comparison of (R)-crizotinib and (S)-crizotinib in these vaccination assays led to the conclusion that the (R) enantiomer is more efficient in inducing ICD than its stereoisomer (Supplementary Fig. 5b–d).

We conclude that (R)-crizotinib acts as a TKI that favors the induction of ICD in a variety of cancer cells, independently of their ALK status.

**Fig. 3** (R)-crizotinib had anti-cancer vaccination effects on NSCLC. Wild type (*WT*) TC1 cells were treated with mitoxantrone (MTX; 4 μM), cisplatin (CDDP; 150 μM), mitomycin C (MitoC; 150 μM) alone or in combination with (R)-crizotinib (Criz, 10 μM) for 24 h. Then the cells were collected and subcutaneously (*s.c.*) injected ($10^6$ cells per mouse) into the left flank of immunocompetent C57BL/6 mice (**a**). PBS was used as control. Two weeks later all mice were rechallenged with living TC1 cells ($2 \times 10^5$ per mouse) in the right flank. The evolution of tumor incidence over time was reported as Kaplan–Meier curves (**b**–**d**). Statistical significance was calculated by means of the Likelihood ratio test. ***$p < 0.001$ compared to PBS group; $n = 10$ per group. Final tumor size distribution at endpoint is shown in **e**. Statistical significance was calculated by means of the ANOVA test for multiple comparisons, ***$p < 0.001$ as compared to the PBS group. **f**, **g** ANXA1-deficient (*Anxa1*$^{-/-}$) or HMGB1-deficient (*Hmgb1*$^{-/-}$) TC1 cells treated 24 h with (R)-crizotinib/ CDDP combination or receiving an equivalent volume of PBS were *s.c.* inoculated in the flank of WT C57BL/6 mice, which were 2 weeks later rechallenged in the opposite flank with living TC1 cells. **h**, **i** *WT* TC1 cells treated 24 h with the combination of (R)-Criz and CDDP were incubated with chicken αCALR antibody or isotype antibody (2.5 μg per $10^6$ cells) for 30 min at room temperature; or an ATP diphosphohydrolase apyrase (5 IU per $10^6$ cells) for 30 min at room temperature before being inoculated *s.c.* into the flank of WT C57BL/6 mice, which were 2 weeks later rechallenged in the opposite flank later with living TC1 cells. Statistical significance was calculated by means of the Likelihood ratio test. ***$p < 0.001$ compared to PBS group; ###$p < 0.001$ as comparing indicated groups, $n =$ minimum of 6 mice per group

**Improvement of chemotherapy by (R)-crizotinib.** Driven by the consideration that (R)-crizotinib induces ICD when combined with CDDP or MitoC, we investigated the therapeutic potential of such a combination regimen in immunocompetent mice. In conditions in which established *s.c.* MCA205 cancers failed to reduce their growth in response to mono-therapy with either intratumoral (*i.t.*) administration of (R)-crizotinib (Supplementary Fig. 6a) or systematic treatment with CDDP (Supplementary Fig. 6b) or MitoC (Supplementary Fig. 6c), the combination therapy of (R)-crizotinib administered *i.t.* (but less so (S)-crizotinib) together with systemic CDDP or MitoC resulted in a significant delay in disease progression (Supplementary Fig. 6b–d). The growth of transplanted MCA205 tumors was also efficiently controlled when either CDDP or MitoC were combined with systemic (intraperitoneal) injections of (R)-crizotinib (Supplementary Fig. 6e, f), which reached a concentration of ~10 μM in plasma and tissues (Supplementary Fig. 6g). None of these anti-neoplastic effects on MCA205 cancers were observed when the tumors evolved in *nu/nu* mice that lack thymus-dependent T lymphocytes (Supplementary Fig. 6h, i). TC1 cancers growing subcutaneously on immunocompetent animals reduced their growth in response to the combinations of these drugs (Supplementary Fig. 6k–m). Moreover, knockout of either *Anxa1* or *Hmgb1* in MCA205 and TC1 cell abolished the tumor-growth reducing effect of the combination therapy with CDDP plus (R)-crizotinib (Supplementary Fig. 6j, m).

Even more spectacular effects were obtained against established orthotopic NSCLC TC1 tumors expressing luciferase to facilitate the monitoring of lung cancers in immunocompetent mice (Fig. 4a, b). In this model, systemic therapy with (R)-crizotinib, CDDP or MitoC alone had no significant therapeutic effects. However, the combination of (R)-crizotinib with CDDP or MitoC caused the (at least temporary) disappearance of more than 50% of the cancers, as well as an increase in overall survival (Fig. 4c–h, Supplementary Fig. 7a). Mice that stayed tumor-free for more than 3 months after the combination treatment with (R)-crizotinib plus CDDP were challenged with both TC1 and MCA205 cells injected into opposite flanks (Fig. 4i). Importantly, those mice that had been cured from established TC1 lung cancers failed to develop *s.c.* TC1 lesions, but did develop MCA205 cancers, which are antigenically different (Fig. 4j, k). As a further proof that the observed therapeutic effects are indeed mediated by the cellular immune system, the best combination therapy (namely, (R)-crizotinib plus CDDP) completely lost efficacy against TC1 lung cancers developing in athymic *nu/nu* mice (Fig. 4l–n, Supplementary Fig. 7b), or when CD4$^+$ or CD8$^+$ T lymphocytes were depleted by the injection of suitable antibodies (Supplementary Fig. 7c–e). Blockade of CD11b partially compromised the efficacy of the treatment with (R)-crizotinib plus CDDP (Supplementary Fig. 7f)

We next investigated the possibility to combine (R)-crizotinib with CDDP for the treatment of oncogene and carcinogen-induced lung cancers. In the model of Kras-activated, Trp53-deleted (KP) lung cancer (in which a Cre-encoding adenovirus is instilled into the trachea of mice bearing a mutant *Kras* antigen downstream of a LOX-Stop-LOX cassette, as well as Cre-excisable *Trp53*)[30], the combination therapy was more efficient in reducing tumor burden than either (R)-crizotinib or CDDP alone (Fig. 5a–d). This applies as well to a model of urethane-induced NSCLC treated from the moment of micro-computed tomography detection (Fig. 5e). Again, the numbers of neoplasias, as well as the total tumor burden were reduced by co-treatment with (R)-crizotinib and CDDP (Fig. 5f–i).

Of note, (R)-crizotinib, alone or in combination with CDDP, increased the infiltration of Kras-induced NSCLC by CD8$^+$ cytotoxic T lymphocytes (CTL), yet had no major effect on the local frequency of Foxp3$^+$ regulatory T cells (Treg), meaning that it improved the CD8/Foxp3 ratio in the tumor bed (Supplementary Fig. 8a–h). Moreover, crizotinib favored the infiltration of cancers by interferon-γ (IFNγ)-producing T lymphocytes in urethane-induced NSCLC (Supplementary Fig. 8i–k). In addition, *s.c.* MCA205 cancers treated with (R)-crizotinib, alone or together with CDDP, exhibited a significant increase ($p < 0.001$, ANOVA test) in the local presence of activated dendritic cells (DC) expressing CD11c and CD86 (Supplementary Fig. 9a, d), CTL (Supplementary Fig. 9b, e). This was accompanied by a (though non-significant) reduction in Treg (Supplementary Fig. 9c, f), translating into a significant increase ($p < 0.001$) in the CD8/ Foxp3 ratio that was more pronounced ($p < 0.01$) for the combination regimen than for (R)-crizotinib alone (Fig. 6a). Cytofluorometric analyses (Supplementary Fig. 10) confirmed crizotinib-induced changes in the immune infiltrate including an increase in inflammatory macrophages (Supplementary Fig. 9g), NKT cells (Supplementary Fig. 9i, j), but no change in the local presence of activated NK cells (Supplementary Fig. 9h), or granulocytic or monocytic myeloid derived suppressor cells (MDSC) (Supplementary Fig. 9k). In addition, (R)-crizotinib tended to increase the frequency of IFNγ producing CD4$^+$ and CD8$^+$ T cells (Fig. 6b, c, Supplementary Fig. 11), as well as that of interleukin-17 (IL-17) producing CD4$^+$ T lymphocytes (Fig. 6d, Supplementary Fig. 11). RNAseq analyses led to the identification of a set of genes that are significantly upregulated by the combination of CDDP and (R)-crizotinib. Among this set, genes that contribute to IFNγ production and its biosynthesis, as well as T cell activation markers were identified by Go term analysis (Fig. 6e, f, Supplementary data 3). Importantly, systemic injection of an IFNγ IFNγ-neutralizing antibody abolished the efficacy of the combination therapy with CDDP plus (R)-crizotinib in all transplantable tumor models investigated here (TC1 or MCA205 tumors, subcutaneous or orthotopic) (Fig. 6g–k). Similarly, the

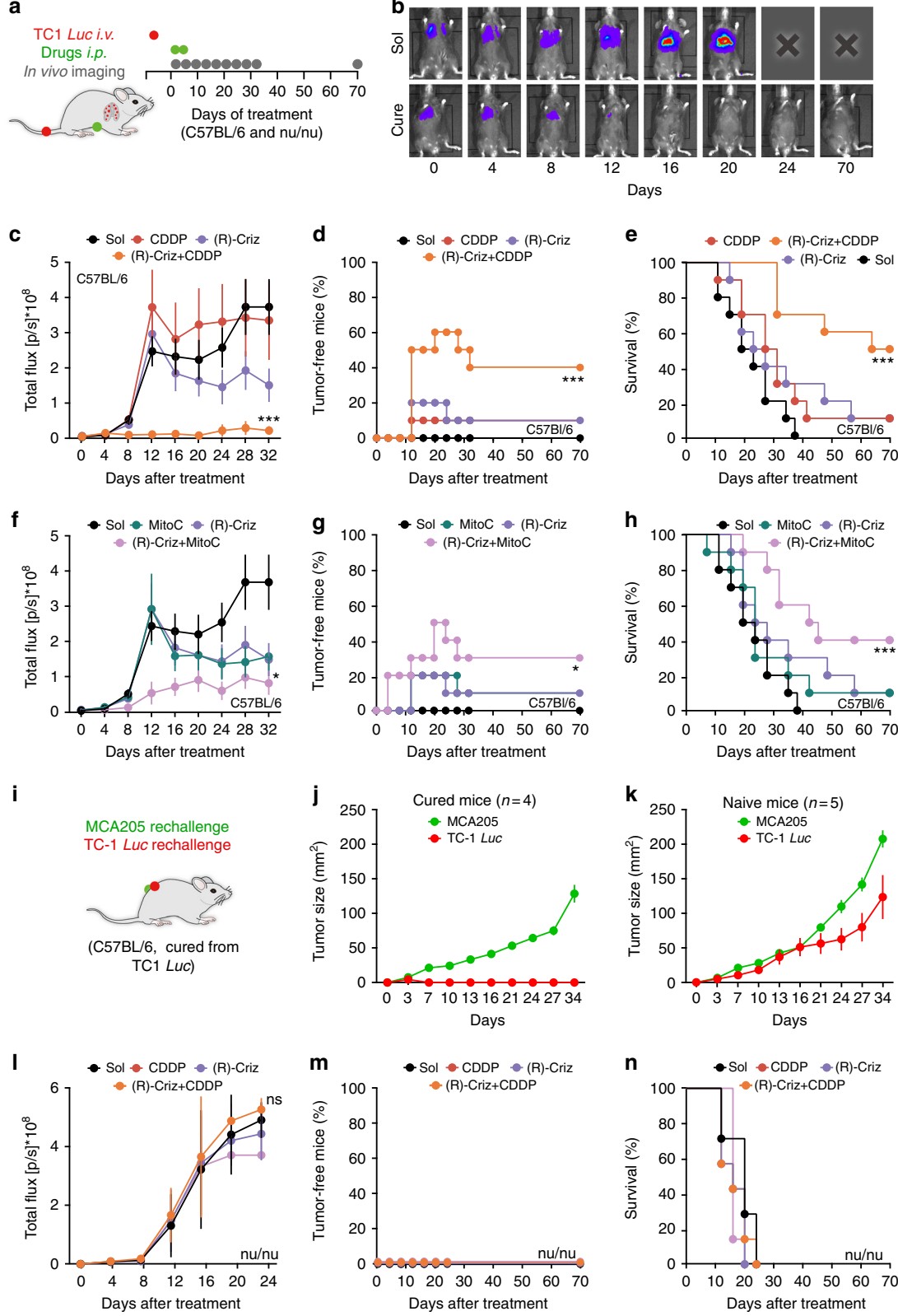

systemic neutralization of the type-1 interferon receptor (IFNAR) abolished the efficacy of the combination therapy against subcutaneous orthotopic cancers (Supplementary Fig. 12a–c). Neutralization of IL-12b failed to affect therapeutic efficacy of the combination therapy (Supplementary Fig. 12d–f), although its RNA level was found significantly upregulated in RNA Seq analysis (Fig. 6e, f) and crizotinib caused a small increase in IL-12

secretion by bone-marrow derived dendritic cells (BMDC) in vitro (Supplementary Fig. 12g–i).

Of note, the (R)-crizotinib effects apparently depend on the tumor microenvironment, because in vitro treatment of naïve CD4[+] T cells in the presence of the TKI and suitable differentiation factors (Supplementary Fig. 13a) failed to affect the expression and secretion of major cytokines (IFNγ, IL-4, IL-9,

**Fig. 4** (R)-crizotinib in combination with CDDP or MitoC had synergistic anti-cancer effects on established orthotopic NSCLC model. Living TC1 cells stably expressing luciferase activity (TC1 *Luc*, $5 \times 10^5$ per mouse) were intravenously (*i.v.*) injected into wild type C57BL/6 mice ($n = 10$ mice per group) (**b-h**) or nude mice (*nu/nu*; $n = 7$ per group) (**l-n**) until tumor incidence in the lung can be detected by bioluminescence. Then mice received repeated treatments with solvent control (Sol), mitoxantrone (MTX; *i.p.*, 5.2 mg Kg$^{-1}$), cisplatin (CDDP; *i.p.*, 0.25 mg Kg$^{-1}$), mitomycin C (MitoC; *i.p.*, 0.25 mg Kg$^{-1}$) alone or in combination with (R)-crizotinib or (S)-crizotinib (Criz, *i.p.*, 50 mg Kg$^{-1}$) at day 0 and day 2. Tumor size was monitored by luciferase activity every 4 days (**a**). Representative time lapse images of a control mouse and a cured animal are reported in **b**; average (mean ± s.e.m.) tumor growth curves are reported in **c**, **f**, **l**; percentage of tumor free mice are reported in **d**, **g**, **m**; overall survival are reported in **e**, **h**, **n**. **i–k** cured immunocompetent mice were rechallenged with living cancer cells of the same type (TC1), as well as different type (MCA205), tumor size was monitored accordingly (**j**). A group of naïve mice were used to confirm the normal growth of both cell types (**k**). Statistical significance was calculated by means of the ANOVA Type 2 (Wald test) for tumor growth curves, or Likelihood ratio test for tumor-free and overall survival curves. *ns*, not significant; \*$p < 0.05$, \*\*\*$p < 0.001$ compared to Sol

IL-17) and Foxp3 by all major CD4$^+$ T T cell subsets (Supplementary Fig. 13b–i). Similarly, systemic treatment of tumor-free mice with (R)-crizotinib, alone or in combination with CDDP (Supplementary Fig. 13j), failed to affect the expression of these factors by splenic CD4$^+$ and CD8$^+$ T lymphocytes (Supplementary Fig. 13k–p), neither affected those factors by lymph nodes' CD4$^+$ and CD8$^+$ T lymphocytes (data not shown).

Altogether, these results indicate that (R)-crizotinib can be combined with chemotherapeutic agents to stimulate IFNγ-dependent anticancer immune responses in various models of NSCLC.

**(R)-crizotinib synergizes with immune checkpoint blockers.** In vitro treatment of several human (A549, HCT116, U2OS) or mouse cancer cell lines (CT26, MCA205, TC1) with crizotinib induced the expression of PD-L1 protein within 24 h (Supplementary Fig. 14). In tumor bearing mice, crizotinib increased the expression of the PD-1 (and LAG-3 but neither CTLA-4 nor TIM-3) on circulating CD4$^+$ or CD8$^+$ T lymphocytes (Supplementary Figs. 15, 16). Treatment of mice with (R)-crizotinib in combination with CDDP caused an increase in the mRNA levels of PD-1, PD-L1, and CTLA-4 within the tumor (Fig. 6l). Moreover, (R)-crizotinib alone or with CDDP induced PD-1 expression on tumor-infiltrating CD4$^+$ Foxp3$^-$ but not in CD4$^+$Foxp3$^+$ (Treg) cells bearing the exhaustion marker ICOS (Fig. 6m, n). As a result, we investigated whether immune checkpoint blockade (ICB) would be able to improve the therapeutic outcome. For this we evaluated all possible combinations of (R)-crizotinib, CDDP-based chemotherapy and dual ICB therapy with PD-1 and CTLA-4-targeting antibodies on *s.c.* MCA205 fibrosarcomas (Supplementary Fig. 17a, b) or *s.c.* TC1 (Fig. 7a). Based on the consideration that ICD induction can sensitize cancer to subsequent ICB treatment[31], ICB was performed after the treatment with (R)-crizotinib and CDDP by three injections of CTLA-4 and PD-1 blocking antibodies, starting from 8 days after the administration of the small molecules (Fig. 7a). While immune checkpoint inhibitors on their own had no therapeutic effects, the combination of ICB with (R)-crizotinib (and optionally with CDDP) improved therapeutic outcome, causing complete regression of > 60% of *s.c.* MCA205 fibrosarcomas (Supplementary Fig. 17a, b) and reducing the growth of *s.c.* TC1 cancer (Fig. 7b, c).

Next, we applied this chemo-immunotherapeutic protocol to established orthotopic TC1 *Luc* NSCLC. The small molecule (R)-crizotinib and/or CDDP were administered upon bioluminescence-based lung cancer diagnosis, followed by the initiation of immunotherapy one week later (Fig. 7d). Cure rates (evaluated as complete disappearance of a diagnostic pulmonary bioluminescence signal) and long-term survival reached > 88% for the triple combination therapy consisting of (R)-crizotinib, CDDP and ICB (Fig. 7e–i). Of note, comparison of different immunotherapies (CTLA-4 blockade alone, PD-1 blockade alone, or combination therapy) led to the conclusion that PD-1 (not

CTLA-4) blockade was sufficient to achieve this high cure rate (Fig. 7j, Supplementary Fig. 17c–f). The combination of (R)-crizotinib, CDDP and PD-1 blockade achieved a 100% cure rate (15 out of 15 mice) in yet another orthotopic NSCLC model, namely Lewis lung carcinoma-1 (LLC1) tumors forming after intrathoracic injection (Supplementary Fig. 17g, h). Only (R)-crizotinib, not (S) crizotinib, was able to sensitize TC1 lung cancers to cure with PD-1 blockade (Supplementary Fig. 17i, j). Mice that had been cured from orthotopic TC1 lung cancers by the combination of (R)-crizotinib, CDDP and PD-1 blockade became resistant against s.c. rechallenge with TC1 tumors (Supplementary Fig. 17k–n).

Cancer patients do not tolerate simultaneous administration of crizotinib and PD-1 blockade due to hepatotoxicity[32]. Similarly, mice receiving simultaneous treatment with such agents exhibited signs of liver toxicity that were not observed when PD-1 blockade was started one week after crizotinib treatment (Supplementary Fig. 18a–d). This correlated with the capacity of (R)-crizotinib (alone or in combination with CDDP) to transiently induce PD-L1 expression in the liver (but not in other tissues) (Supplementary Fig. 18e–h).

Altogether, these results demonstrate that (R)-crizotinib can be useful in boosting anticancer chemotherapy effects, as well as subsequent responses to ICB-based immunotherapy.

**Discussion**
The incidence of NSCLC treated with crizotinib is rather low, because only 1% of these cancers have *ROS1* rearrangements[33] and only 4–5% harbor *ALK* genetic rearrangements[34]. Such NSCLC patients, who typically are non-smokers with adenocarcinomas, have been treated with crizotinib in first line (for ALK-positive cancers only) and, once the tumors become crizotinib-resistant and relapse, with other TKI inhibitors such as ceritinib[35,36]. Typically, crizotinib resistance is accompanied by secondary mutation of *ALK*, amplification of the *ALK* fusion gene or aberrant activation of other oncogenic kinases such as KIT or EGFR[37]. Although crizotinib mediates occasional effects against NSCLC bearing activating MET mutations, it is generally not efficient for the treatment of NSCLC which lacks oncogenic kinases that might be inhibited by crizotinib[38]. Hence the available evidence suggests that crizotinib is acting on-target, by inhibiting ALK, ROS1 or MET to mediate its antineoplastic effects.

Here, we propose that crizotinib also has an off-target effect that becomes manifest in the context of cisplatin-based chemotherapy and that may be taken advantage of to treat NSCLCs that bear none of the canonical crizotinib targets (Supplementary Fig. 19, graphical abstract). Indeed, when combined with CDDP (or another chemotherapeutic agent such as mitomycin C), crizotinib is effective against s.c. and orthotopic TC1 or LLC1 NSCLC, as well as against more realistic NSCLC models that were either induced by chemical carcinogenesis or by the conditional activation of mutant Kras and deleted Trp53. In all these models, crizotinib combined with CDDP caused signs of an anticancer

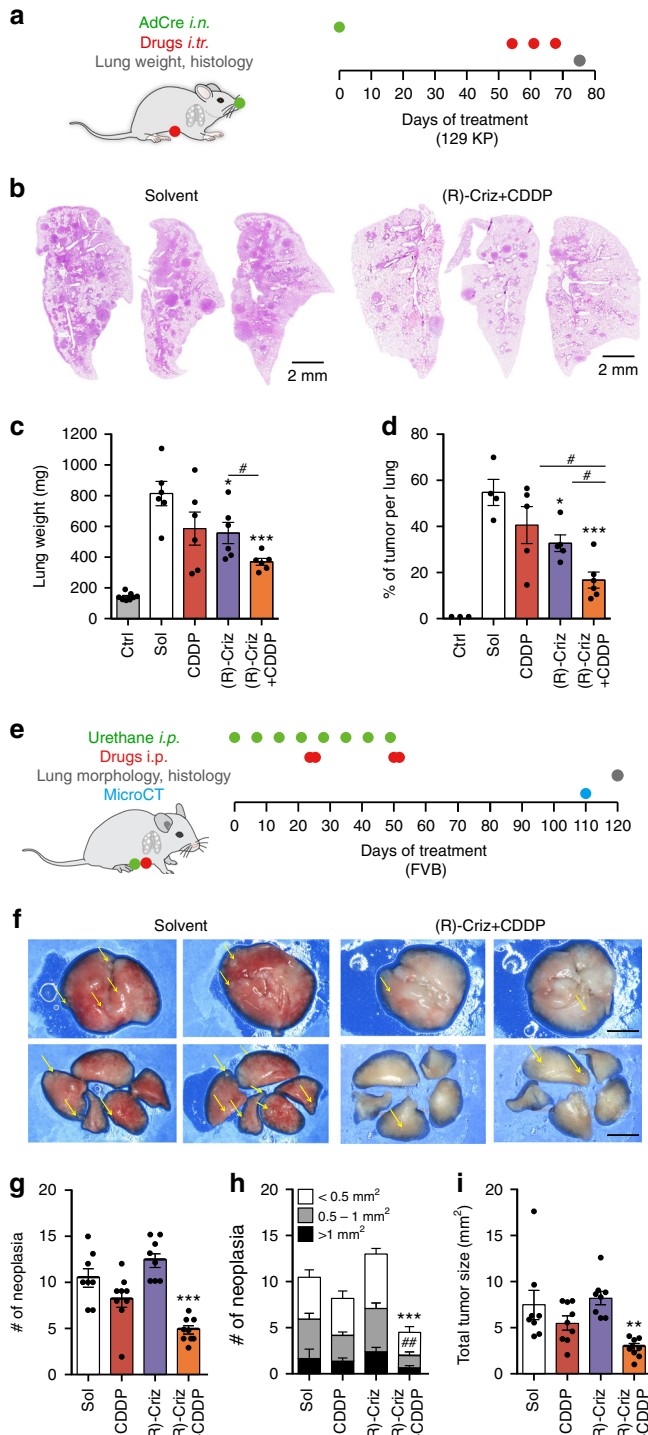

**Fig. 5** (R)-crizotinib in combination with CDDP had synergistic antitumor effect on oncogene and carcinogen-induced lung cancer models. **a–d** Kras[LSL−G12D/+]; Trp53[flox/flox] (KP) mice were used as a conditional mouse model of NSCLC, tumor initiation and drug treatments were performed according to the scheme demonstrated in **a**. At the end of the experiment, all mice were euthanized and lungs were sampled to measure weight and subjected to further hematoxylin-eosin staining and immunohistochemistry. Representative hematoxylin-eosin stained lung lobes of KP mice that were treated with either solvent (Sol) or the combination of (R)-crizotinib (Criz) with cisplatin (CDDP) are shown in **b**. Lung weight was used as proxy of tumor burden and is depicted for tumor-free (Ctrl) and treated tumor-bearing mice (**c**). Hematoxylin-eosin based tumor area quantification on lung lobe sections of KP mice is presented (**d**). Results are expressed as mean ± s.e.m. *$p < 0.05$; ***$p < 0.001$ as compared to Sol, #$p < 0.05$ as comparing indicated groups using Student's t-test, n = minimum of 6 mice per group. Friend Virus B (FVB) mice were used as a spontaneous mouse model of lung adenocarcinoma, tumor induction and drug treatments were performed according to the scheme demonstrated in **e**. Tumor development was monitored with microCT photographing and all mice were euthanized to obtain lung lobes for further histology. Representative stereo microscope scans of lung lobes (**f**), as well as quantified neoplasia numbers (**g**, **h**) and tumor sizes (**i**) are reported. Results are expressed as mean ± s.e.m. **$p < 0.01$; ***$p < 0.001$ as compared to Sol, ##$p < 0.01$ as comparing neoplasia numbers in specific size using Student's t-test, scale bar equals 1 mm, n = minimum of 9 mice per group

alone), allowing to induce a durable immune response, as well as complete cure against the majority of established orthotopic NSCLC.

As to the mechanisms through which crizotinib favors the induction of anticancer immune responses, it appears that (R)-crizotinib (the TKI) is more efficient than its stereoisomer (S)-crizotinib (the inhibitor of nudix hydrolase 1, NUDT1, a nucleotide pool sanitizing enzyme)[39]. By virtue of simultaneous inhibiting several tyrosine kinases (namely ALK, ROS1, MET, and JAK2), (R)-crizotinib combined with CDDP (or mitomycin C) induces all the molecular and functional hallmarks of ICD in a variety of human and mouse NSCLC cell lines, even if such cells lack activating mutations of ALK or ROS1. This is specific for the drug combination, meaning that crizotinib, CDDP or mitomycin C do not induce ICD on their own; this effect was only obtained when crizotinib was combined with the chemotherapeutics. Removal of any of the known DAMPs that are required for ICD including ANXA1, ATP, CALR, or HMGB1 abolished the capacity of crizotinib plus CDDP-treated NSCLC cells to elicit an anticancer immune response in vaccination assays. Hence, it appears that the combination treatment induces a canonical ICD pathway that resembles that induced by oxaliplatin or anthracyclines[17].

In the clinical trials that led to the approval of crizotinib for the treatment of a subcategory of NSCLC bearing ALK and ROS1 activating chromosome translocations, the drug was usually compared to, but not combined with, CDDP[35,38]. Hence, there is no information on the possible benefits of the combination therapy in NSCLC. Only a minority of NSCLC with ALK rearrangements (ALK+) that had been treated with crizotinib responded to PD-1 blockade[40,41]. Preclinical evidence indicates that ICD inducing cytotoxicants (such as oxaliplatin and cyclophosphamide, especially if combined) can sensitize NSCLC to CTLA-4/PD-1 blockade[31]. Importantly, nivolumab was particularly efficient among those NSCLC patients who demonstrated a response to first-line chemotherapy[42], suggesting that this may apply to patients as well. Moreover, distinct chemotherapeutic regimens can be advantageously combined with ICBs for the

immune response, as indicated by an increase in the T cell infiltrate of the tumors. Removal of T lymphocytes from the system curtailed the efficacy of the combination regimen. Analysis of the immune infiltrate pleaded in favor of an improvement of the local immune tonus (as indicated by an increase in the ratio of CD8+ CTL over Foxp3+ Tregs), as well as an increased production of IFNα by tumor-infiltrating T cells. Indeed, neutralization of IFNγ was sufficient to abolish the beneficial effects of the combination therapy with crizotinib and CDDP. More importantly, the combined action of crizotinib and CDDP led to the intratumoral upregulation of CTLA-4, PD-1, and PD-L1. Indeed, crizotinib plus CDDP sensitized tumors to immunotherapy with antibodies blocking CTLA-4 and PD-1 (or PD-1

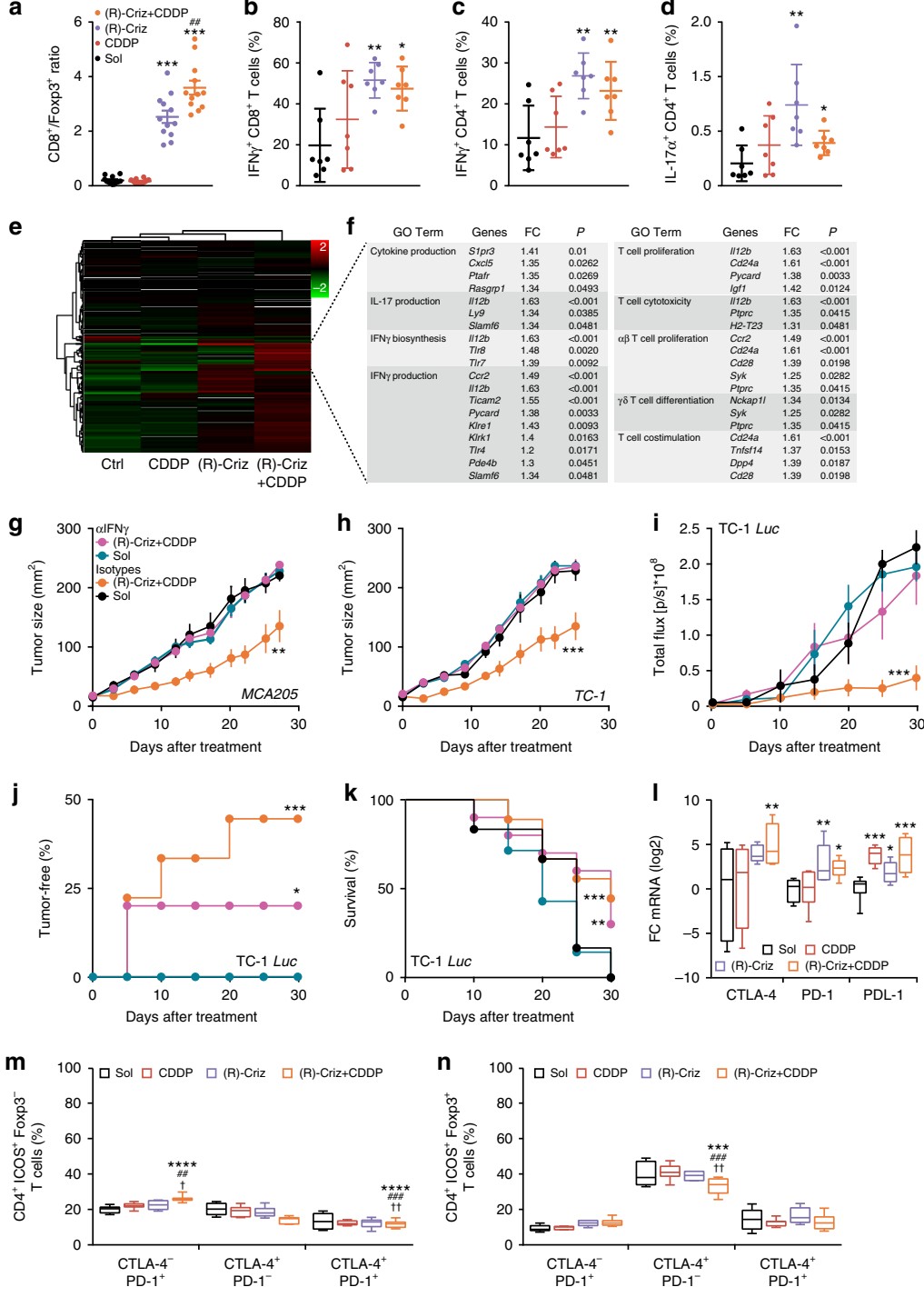

treatment of metastatic NSCLC[43,44]. Based on these premises, it might be attempted to extrapolate the results obtained in the mouse models described here to design a clinical trial in which NSCLC patients would receive standard chemotherapy in combination with crizotinib (and this independently of the status of *ALK* and *ROS1*), followed by ICB. Of note, in mice crizotinib could be safely administered without any obvious toxicity to reach a concentration of 10 μM in plasma, lung tissues and tumors, which is the dose that is effectively inducing ICD hallmarks in cancer cells that lack *ALK* rearrangements. Moreover, no toxicity was observed when crizotinib and chemotherapy were given first and immunotherapy was administered one week later

when crizotinib levels had declined below detection limits. Accordingly, *PD-L1* mRNA expression increased in the liver 2 days after crizotinib administration, yet returned to basal conditions one week later. Hence, a sequential regimen may avoid the toxicity that has led to the premature termination of a clinical trial in which crizotinib was co-administered with the PD-1 targeting antibody nivolumab[32].

## Methods

**Cell culture conditions and chemical products.** Unless specified, all wild-type cancer cell lines were purchased from the American Type Culture Collection (ATCC, Rockefeller, MD, USA) and were maintained at 37 °C under 5% $CO_2$ in

**Fig. 6** (R)-crizotinib induces immune infiltration in established tumors. MCA205 tumors-bearing mice received injections (*i.p.*) of solvent (Sol), cisplatin (CDDP), (R)-crizotinib (Criz) or the combination at day 0 (when tumors became palpable) and day 2. Samples were harvested at day 8 for immunofluorescence staining (**a**); or RNA extraction for qPCR analysis using primers for *CTLA-4*, *PD-1* and *PD-L1* (**l**), $n = 12$ per group. Alternatively tumors were harvested at day 10 for flow cytometry ($n = 7$ per group) (**b–d**, **m–o**), or RNA extraction and RNASeq analysis ($n = 4$ per group) (**e**, **f**). **a** CD8$^+$/Foxp3$^+$ ratios are depicted as mean ± s.e.m. ***$p < 0.001$ as compared to Sol, ##$p < 0.01$ as comparing indicated groups using Student's *t*-test. (**b–d**) IFNγ-producing and IL-17-producing T cells were quantified by flow cytometry and are reported as dot plots. Statistical significance was calculated using two-way ANOVA test, *$p < 0.05$; **$p < 0.01$; as compared to Sol. **e**, **f** GO term enrichment analysis for T cell activation related genes are depicted as fold change (FC) with adjusted *P* values (*P*) for the condition Sol vs. (R)-Criz + CDDP (**f**). WT C57BL/6 mice bearing subcutaneous (*s.c.*) MC205 (**g**), *s.c.* TC1 (**h**), or orthotopic TC1 *Luc* tumors (**i–k**) were treated with Sol or (R)-Criz + CDDP, accompanied by systemic administration of an IFNγ neutralizing antibody (αIFNγ) or or isotype antibody (αIso). Statistical significance was calculated by ANOVA Type 2 (Wald test) for tumor growth, or Likelihood ratio test for tumor incidence and overall survival. *ns*, not significant; *$p < 0.05$, **$p < 0.01$, ***$p < 0.001$ compared to Sol. **l** qPCR analysis of *CTLA-4*, *PD-1* and *PD-L1* mRNA in MCA205 tumors. Expressions of immune checkpoint markers on tumor infiltrating CD4$^+$FOXP3$^-$ICOS$^+$ (**m**) or CD4$^+$FOXP3$^+$ICOS$^+$ (**n**) T cells were quantified by flow cytometry. The frequency of CTLA-4$^+$PD-1$^-$, CTLA-4$^-$PD-1$^+$ or CTLA-4$^+$PD-1$^+$ cells was determined among the indicated subsets. Statistical significance was calculated by two-way ANOVA test for multiple comparisons, *$p < 0.05$; **$p < 0.01$; ***$p < 0.001$ as compared to Sol; ##$p < 0.01$; ###$p < 0.001$ as compared to CDDP; †$p < 0.01$; ††$p < 0.001$ as compared to (R)-Criz

media supplemented with 10% (v/v) fetal bovine serum (FBS), 10 U mL$^{-1}$ penicillin sodium and 10 µg mL$^{-1}$ streptomycin sulfate. Human osteosarcoma U2OS cells, their derivatives expressing GFP-LC3, CALR-RFP or HMGB1-GFP, human cervical adenocarcinoma HeLa cells and murine methylcholanthrene induced fibrosarcoma MCA205 cells were cultured in Dulbecco's modified eagle medium (DMEM); murine non-small-cell lung carcinoma (NSCLC) TC1 cells, LLC1 cells, as well as their derivatives expressing firefly luciferase (*Luc*), human NSCLC cell lines NCI-H2228 and NCI-H1650 were cultured in Roswell Park Memorial Institute (RPMI) 1640 medium; human colon adenocarcinoma HCT-116 cells were cultured in McCoy's 5 A medium. AnnexinA1-deficient (*Anxa1$^{-/-}$*) MCA205 and TC1 cell lines were generated by means of the CompoZr® Zinc Finger Nuclease Technology (CKOZFN26764, Sigma-Aldrich, St Louis, MI, USA), as per manufacturer's recommendations[20]; HMGB1-deficient (*Hmgb1$^{-/-}$*) MCA205 and TC1 cell lines were generated by means of the CRISPR/Cas-mediated genome editing (sc-420871-KO-2, Santa Cruz Biotechnology), as per manufacturer's recommendations. All culture media and supplements were bought from Gibco (Carlsbad, CA, USA) and plastic materials came from Corning (Corning, NY, USA). The Published Kinase Inhibitor Set (PKIS) and the complementary set, PKIS2 were obtained from SGC-UNC (Structural Genomics Consortium-UNC Eshelman School of Pharmacy at the University of North Carolina, Raleigh-Durham, NC, USA) and detailed information regarding chemical structures, kinase inhibition potency and kinase specificity can be found at ChEMBL (http://chembl.blogspot.co.uk/2013/05/pkis-data-in-chembl.html). The collection of 30 tyrosine kinase inhibitors was purchased from LC Laboratory (Woburn, MA, USA). Additionally used (R)-crizotinib, (S)-crizotinib, cisplatin (CDDP), mitomycin C (MitoC), mitoxantrone (MTX), staurosporine (STS), urethane, PEG400, TWEEN80 and DMSO, as well as other unspecified chemicals and salts were purchased from Sigma-Aldrich.

**High throughput screening.** Human osteosarcoma U2OS cells stably expressing GFP-LC3 or HMGB1-GFP/CALR-RFP[22] were seeded in 384-well black imaging plates (Greiner-bio-one, Kremsmünster, Austria) at a density of 1500 cells per well and allowed to adapt for 24 h. The following day, compounds from the PKIS were added at a final concentration of 10 µM, or the 30 tyrosine kinase inhibitors were added at final concentrations of 5 and 10 µM. Two micrometer MTX was used as a positive control. Cells were incubated for 8 h, 12 h, or 32 h for the detection of CALR redistribution, LC3 accumulation or HMGB1 release respectively. After the indicated time, plates were fixed with 4% paraformaldehyde (PFA) in PBS containing 2 µg mL$^{-1}$ Hoechst 33342 (Thermo Scientific, Waltham, MA, USA) overnight at 4 °C. After 3 additional washing steps plates were filled with 50 µL PBS per well and subjected to automated image acquisition and subsequent image analysis. For automated fluorescence microscopy, a robot-assisted Molecular Devices IXM XL BioImager (Molecular Devices, Sunnyvale, CA, USA) equipped with a Sola light source (Lumencor, Beaverton, OR, USA), adequate excitation and emission filters (Semrock) a 16-bit monochrome sCMOS PCO.edge 5.5 camera (PCO, Kelheim, Germany) and a 20X PlanAPO objective (Nikon, Tokyo, Japan) was used to acquire a minimum of 4 view fields of each well. Following images were processed and segmented with the MetaXpress software (Molecular Devices) to analyze GFP-LC3 granularity, CALR-RFP granularity at the cellular membrane, the decrease of nuclear HMGB1-GFP intensity, as well as nuclear shrinkage to indicate apoptotic cell death. Data mining was conducted using the freely available software R (https://www.r-project.org). Data was intra-plate normalized as the ratio to plate means for each data point, while Z-scores were employed for inter-plate normalization.

**In vitro validation of CALR exposure, ATP release and HMGB1 release.** U2OS, MCA205 or TC1 cells were seeded in 6 well plates and allowed to adapt for 24 h. Following, cells were treated with MTX (2 µM), CDDP (150 µM), MitoC (150 µM), (R) or (S)-crizotinib (1, 5, or 10 µM), alone or in combinations, for 6 h. Following the cells were washed and fresh medium was added for additional 24 h. Cells were

collected and washed for staining with an anti-CALR antibody (ab2907, Abcam) diluted 1:100 in flow cytometry buffer (1% BSA in PBS) for 45 min. Then the cells were washed and incubated with an Alexa Fluor 488®-conjugated goat-anti-rabbit antibody (Invitrogen, Carlsbad, CA, USA) diluted 1:250 in flow cytometry buffer for 30 min. Cells were washed and resuspended in flow cytometry buffer containing 2 µg mL$^{-1}$ 4′,6-diamidino-2-phenylindole (DAPI) and then immediately subjected to flow cytometry (BD LSRFortessa, BD Biosciences, San José, CA, USA). Data were further processed with the FlowJo software (Tree Star, Inc., Ashland, OR, USA) to assess the percentage of CALR$^+$ DAPI$^-$ cells. ATP concentrations in the supernatant of cells upon the indicated treatment were measured by means of an ENLITEN ATP Assay kit (Promega, Fitchburg, WI, USA), based on the ATP dependent luciferin–conversion, that yields detectable bioluminescence, according to the manufacturer's protocol. HMGB1 concentrations in the supernatant of cells following the indicated treatment were measured by means of an enzyme-linked immunosorbent assay (ELISA) kit (Shino test corporation, Tokyo, Japan), according to the manufacturer's protocol. Luminescence and absorbance were measured using a SpectraMax I3 multi-mode microplate reader (Molecular Devices).

**Clonogenic assay.** MCA205 or TC1 cells were seeded in 6-well plates (200 cells per well in 2 mL medium) and let adapt for 24 h before the treatment with different concentrations of (R)/(S)-crizotinib, alone or in combination with CDDP. Another 24 h after the treatment, medium was replaced, and the plates were incubated for additional 12 days before fixation and crystal violet staining (0.05% crystal violet, 1% PFA, 1% methanol in PBS). The plates were scanned and the area of clones was quantified with the free software image J (https://imagej.nih.gov/ij/).

**Quantitative RT-PCR.** Total RNA extraction of cells was performed with the GeneJET RNA Purification Kit (Life Technologies). Total RNA extraction of established tumors was performed with the RNeasy Plus Mini Kit (Qiagen, Hilden, Germany) and RNAlater RNA Stabilization Reagent (Qiagen) following the manufacturers' instructions. Total RNA (2.5 µg from each sample) was then reverse transcribed into cDNA with the Maxima First Strand cDNA Synthesis Kit (Life Technologies). Expression of the interested genes was analyzed by means of SYBR® Green based Quantitative PCR using the Power SYBR™ Green PCR Master Mix in StepOnePlus Real-Time PCR System (Applied Biosystems, Foster City, CA, USA). qRT-PCR data were normalized to the expression levels of the housekeeping gene hypoxanthine phosphoribosyltransferase 1 (*HPRT1*). Information of PCR primers is depicted in Supplementary table 1.

**Quantification of cell death by flow cytometer.** Cell death was assessed by means of the FITC-Annexin V detection kit I (BD Biosciences) following the manufacture's procedures. Briefly, cells treated in 6-well plates were collected and washed in PBS before the cell pellet was resuspended in 50 µL staining buffer containing FITC-conjugated AnnexinV antibody. Samples were then incubated in the dark for 15 min before adding 400 µL staining buffer supplemented with 2 µg mL$^{-1}$ DAPI. Acquisitions were performed on a CyAn ADP cytofluorometer (Beckman Coulter, Indianapolis, USA), and data were statistically evaluated using FlowJo.

**In vitro analysis of PD-L1 and MHC Class I/II expression by flow cytometry.** Cells were seeded in 96-well plates (10,000 cells per well for murine cells) in 100 µL growth medium and let adapt for 24 h before treatment. Cells were then treated by different concentrations of crizotinib, alone or in combination with cisplatin, for another 24 h, recombinant murine IFNα (2000 IU) or IFNγ (100 ng mL$^{-1}$) was used as positive controls. Post-treated cells were collected in 96-well V-shape plates (Greiner-bio-one). Cells were first stained with Zombie UV dye (Biolegend) for 15 min at 4 °C in the dark, then washed and stained with an anti-mouse CD274 (PD-L1, clone MIH1, eBioscience), anti-mouse MHC Class I (H-2K$^b$, clone

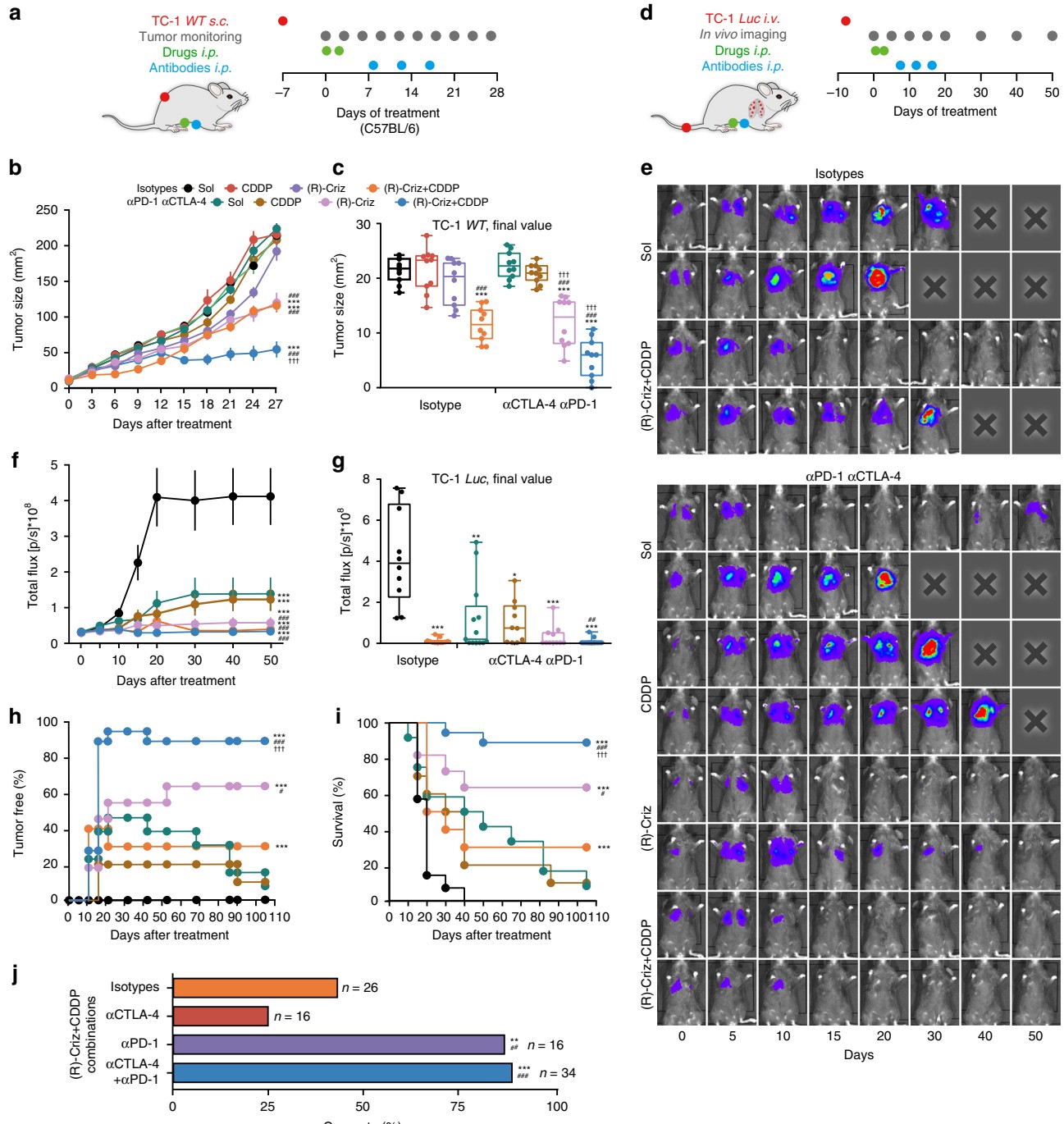

**Fig. 7** Immunogenic chemotherapy of (R)-crizotinib sensitizes NSCLC tumors to checkpoint blockades. **a–c** Treatment of subcutaneous (s.c.) TC1 tumors with injections of solvent control, CDDP alone or in combination with (R)-crizotinib. Isotype monocloncal antibodies (mAbs) or anti-PD-1 mAbs (αpd-1) combined with anti-CTLA-4 mAbs (αCTLA-4) were injected on day 8, 12, and 16 (schedule in **a**). Tumor growth was monitored (**b**, **c**) and expressed as surface (mean ± s.e.m., **b**) or size at endpoint (box plot, **c**). **d–j** Treatment of orthotopic TC1 *Luc* tumors. Once the presence of lung cancer could be detected by bioluminescence (day 0), the animals were treated according to the scheme (**d**). Representative images of tumor development are shown in **e**; average (mean ± s.e.m.) bioluminescence signals are reported in **f**; final values at endpoint are reported as box plot in **g**. The percentage of tumor free mice is reported in **h**; overall survival is reported in **i**. Data in **e–i** include dual checkpoint blockade (αCTLA-4/αPD-1). The effects of dual as compared to single checkpoint blockade at day 70 are shown in **j**. Statistical significance was calculated by means of the ANOVA Type 2 (Wald test) (**b**, **f**), ANOVA test for multiple comparisons (**c**, **g**), Likelihood ratio test (**h**, **i**) or χ² test (**j**). *p < 0.05, **p < 0.01, ***p < 0.001 as compared to control group with isotype; #p < 0.05, ##p < 0.01, ###p < 0.001 as compared to control group with the combination of αPD-1 and αCTLA-4; †††p < 0.001 comparing isotype and αCTLA-4 and αPD-1 combination, n = minimum of 10 animals per group

AF6-88.5.5.3, eBioscience), or anti-mouse MHC Class II (I-A/I-E, clone M5/114.15.2, eBioscience) monoclonal antibody for another 25 min at 4 °C in the dark. After staining, cells were fixed with 4% PFA and kept at 4 °C. Flow cytometry acquisition was performed on a MACSQuant (Miltenyi Biotec), and data analysis was conducted with the FlowJo software.

**In vitro assessment of Il-12 secretion from bone marrow-derived dendritic cells (BMDC)**. Bone marrow cells were flushed with the tibia and femur of C57BL/6 mice using PBS supplemented with 5% FBS. Following centrifugation, cells were resuspended in erythrocytes lysis buffer and passed through 70 μm cell strainer to generate a single cell suspension. Cells were diluted at $1 \times 10^6$ cells mL$^{-1}$ and cultured in DC medium (RPMI-1640 supplemented with 10% FBS, sodium pyruvate, 50 μM 2-mercaptoethanol (Sigma), 10 mM HEPES, and penicillin/streptomycin) containing 40 ng/ml recombinant mouse GM-CSF (PeproTECH) in 6-well plates. Fresh medium was added at days 3 and 6. BMDCs were harvested at day 7, seeded in 96-well plates ($2 \times 10^5$ per well) and let adapt for 24 h. Then the medium was changed to fresh DC medium containing the desired treatment. LPS (eBioscience™, 2.5 μg mL$^{-1}$) was used as a positive control. At the end of the treatment, supernatants were collected for ELISA (REF 433605, Biolegend) analysis of extracellular IL-12 p70.

**Immunoblotting**. Immunoblotting was performed according to standard procedures of the NuPAGE® electrophoresis system (Invitrogen). In brief, protein extracts were obtained by lysing cells in radioimmunoprecipitation assay (RIPA) buffer containing protease inhibitors cocktail, which were separated on 4–12% NuPAGE® Novex® Bis-Tris gels in the NuPAGE® MES SDS Running Buffer and electrotransferred to 0.45 μm polyvinylidene fluoride (PVDF) membranes (Bio-Rad, Hercules, CA, USA) in the Tris-Glycine (TG) buffer. Membranes were incubated for 1 h in 5% non-fat powdered milk dissolved in TBST (Tris-buffered saline containing 0.05% Tween 20) to saturate unspecific binding sites, followed by an overnight incubation (4 °C) with primary antibodies specific for p-EIF2α, ab32157, EIF2α, ab5369, ATF4, ab31390, XBP1, ab37152, β-Actin, ab20272 (Abcam, Cambridge, UK); Hexokinase II, 2867 (Cell Signaling Technology, Danvers, USA). Membranes were washed 5× times with TBST and revealed with suitable horseradish peroxidase-conjugated secondary antibodies (Southern Biotechnologies, Birmingham, AL, USA) for another 2 h at room temperature, followed by additional washing steps and chemiluminescence-based detection with the Amersham ECL Prime detection reagent kit and the ImageQuant LAS 4000 software-assisted imager (GE Healthcare, Piscataway, NJ, USA).

**Immunofluorescence of cells**. Cells were seeded into 384 wells black microplates and treated as indicated. After washing twice with PBS, plates were fixed with 4% PFA/PBS containing Hoechst 33342 (2 μg mL$^{-1}$) for 10 min, washed twice with PBS and incubated with quenching solution (2.67 g NH$_4$Cl in 1 L PBS, pH 7.4) for 5 min. Then cells were washed with PBS and permeabilized with 0.1% Triton-X 100 for 10 min, and rinsed twice with PBS. Plates were blocked with 2% BSA in PBST (PBS, 0.01% Tween-20; v:v) containing 22.52 mg mL$^{-1}$ glycine for 30 min before incubation with primary antibodies overnight at 4 °C. Then the cells were rinsed twice with PBS and incubated with secondary antibodies for 30 min at room temperature in the dark. After additional washing steps, 50 μL PBS were added to each well. The cells were subjected to automated image acquisition and subsequent image analysis as described in the *High throughput screening* part.

**RNA interference in cell culture**. Small interfering RNA (siRNA) sequences targeting ALK, JAK2, MET, ROS1, as well as Non-Targeting siRNA Control pools were obtained from GE-Dharmacon (GE Healthcare, Piscataway, NJ, USA) as a pool of 4 siRNAs (SMARTpool) or aliquots of all 4 individual siRNAs targeting a single gene (Set of 4 Upgrade). SiRNA transfection was performed with the DharmaFECT Transfection Reagent (GE Healthcare) strictly following the manufacture's protocol. Thirty-six hours after transfection, the supernatant was replaced with fresh medium and the cells were incubated for additional 12 h before treatment. Upon treatment the cells were subjected to total RNA extraction for qRT-PCR analysis with primers depicted in Supplementary table 1 and protein extraction for western blotting.

**Assessment of glycolytic flux and mitochondrial respiration**. Flux analysis was employed to measure extracellular acidification rate (ECAR) as an indicator of glycolysis-dependent lactate secretion, and oxygen consumption rate (OCR) as an indicator of mitochondrial respiration, which were performed on a Seahorse XFe96 Analyzer (Seahorse Bioscience, Billerica, MA, USA) as described[45]. Briefly, H1650 (20,000 cells per well) or H2228 (30,000 cells per well) cells were seeded in XF96 Polystyrene Cell Culture Microplates (Seahorse Bioscience) in 80 μL complete culture medium, and allowed to adapt for 24 h before treatment with different concentrations of (R)/(S)-crizotinib for 6 h. Utility Plates were filled with 200 μL of XF Calibrant (Seahorse Bioscience) and the Sensor Cartridges were put on the Utility Plates with all sensors fully immersed in XF Calibrant which were incubated overnight at 37 °C. The following day, medium was replaced with 200 μL/well assay medium (XF-DMEM containing 1 mM L-glutamine, pH to 7.4) and the plate was placed at 37 °C for 60 min. Glucose (90 mM, 25 μL per well), oligomycin (15 μM,

25 μL per well) and 2-deoxyglucose (500 mM, 25 μL per well) were loaded into the Sensor Cartridge for ECAR assessment, while oligomycin (15 μM), CCCP (9 μM), and rotenone (12.5 μM) were loaded for OCR assessment on the Seahorse XFe96 Analyzer. ECAR and OCR were analyzed by the Wave Desktop software (Seahorse Bioscience) and reported as step line chart or histogram (mean ± s.d.).

**Pharmacokinetic of (R)-crizotinib kinetics in murine plasma**. Female C57Bl/6 mice (8 weeks) received a single *i.p.* injection of (R)-crizotinib (50 mg Kg$^{-1}$, dissolved as described above) before blood sampling via cardiac puncture. Blood was collect in BD Microtainer™ Capillary Blood Collector (BD bioscience) containing lithium heparin and was spun to obtain plasma. Plasma was mixed with extraction buffer (methanol, H$_2$O, chloroform; 9:1:1, v:v) at a ratio of 1:10 (v:v) vortexed and centrifugated (10 min at 15,000 × g, 4 °C). Then extracts from the supernatant were dried by evaporation at 40 °C in a pneumatically assisted concentrator (Techne DB3, Staffordshire, UK). The dried extract was re-solubilized with MilliQ water for analysis by liquid chromatography (LC) mass spectrometer (MS). The concentration of crizotinib was calculated based on a crizotinib calibration curve (lower limit of quantification (LLOQ, fixed at 0.0222 μM) and above upper limit of quantification (ULOQ, fixed at 44.4 μM)) using the area of ion signal. To reduce effects of sample carryover, blank samples (water) were injected between biological samples as a rinse step. Targeted analysis was performed using a RRLC 1260 system (Agilent Technologies, Waldbronn, Germany) coupled to a 6500 + QTRAP MS (Sciex, Darmstadt, Germany) equipped with an electrospray ion source. The instruments were operated using a multiple reaction monitoring (MRM) in positive ion mode with a unit resolution for both Q1 and Q3. The optimized MS/MS conditions were: ion spray source temperature at 450 °C, curtain (CUR) gas pressure at 30 psi, gas 1 (GS1) pressure at 50.0 psi and gas 2 (GS2) pressure at 70.0 psi. Ion-spray voltage (IS) was set at 4500 V, collision-activated dissociation (CAD) at High, entrance potential (EP) at 10.0 V and declustering potential (DP) at 96 V. For crizotinib, the collision energy (CE) was set at 39.0 V, and collision exit potential (CXP) at 12 V. For selumetinib ISTD, which was used as an internal standard, the collision energy (CE) was set at 37.0 V, and collision exit potential (CXP) at 18 V. The transitions monitored for quantitation were m/z 450.1 > 260.1 for crizotinib and m/z 457.1 > 361.1 for the selumetinib ISTD. A volume of 2.5 μL of sample were injected on a column Kinetex Polar C18 (50 mm × 2.1 mm particle size 2.6 μm) from Phenomenex, protected by a guard column C18 (5 mm × 2.1 mm) and heated at 40 °C by a Pelletier oven. The gradient mobile phase consisted of water with 0.2% acetic acid (A) and Methanol (B). The flow rate was set to 0.4 mL min$^{-1}$. Initial condition was 95% phase A and 5% phase B. Molecules were then eluted using a gradient from 5 to 95% phase B in 3 min. The column was washed using 95% mobile phase B for 2 mins and equilibrated using 5% mobile phase B for 2 min. The autosampler was kept at 4 °C. The Analyst (Version 1.7) software was used to operate the mass spectrometer. Peak detection, integration and quantification of the analytes were performed using the MultiQuant quantitative software (Version 3.0.3).

**Antitumor vaccination assays and chemotherapy studies with established cancer models**. All mice were maintained in the animal facilities of Gustave Roussy Campus Cancer in a specific pathogen–free, temperature-controlled environment with 12 h light/dark cycles and received food and water ad libitum. All animal experiments were performed in compliance with the EU Directive 63/2010 and specific ethic protocol (Protocols 2016_082 that was approved by the Ethical Committee of the Gustave Roussy Campus Cancer, CEEA IRCIV/IGR no. 26, registered at the French Ministry of Research) Six- to eight-week-old female wild-type C57BL/6 mice were obtained from ENVIGO France (Gannat, France). Six to seven-week-old female athymic nude (*nu/nu*) mice were obtained from Harlan France (Gannat, France).

For antitumor vaccination experiments, wild type MCA205, TC1, as well as their *Anxa1*$^{-/-}$ or *Hmgb1*$^{-/-}$ derivatives were treated with MTX (4 μM), CDDP (150 μM), MitoC (150 μM), (R)-crizotinib (10 μM), (S)-crizotinib (10 μM) alone or in combinations for 24 h in 175 cm$^2$ flasks. Then supernatants and detached cells were collected in a 50 mL falcon tube. The cells were centrifuged and the pellet was further washed with ice cold PBS before resuspending the cells at a concentration of $1 \times 10^7$ cells per mL. When applicable, cells were further incubated with chicken-anti-CALR or isotype antibodies (2.5 μg per $10^6$ cells) for 30 min at room temperature or treated with apyrase (5 IU per $10^6$ cells) for 30 min at room temperature. For vaccination 100 μL of the cellular suspension was *s.c.* injected to the left flank of immunocompetent C57BL/6 mice (8-weeks-old female, $1 \times 10^6$ cells per mouse). PBS was injected as a negative control. Two weeks later, all mice were confirmed tumor-free in the vaccination flank and living cancer cells of the same types ($1 \times 10^5$ cell per mouse for MCA205 cells and its derivatives, $2 \times 10^5$ per mouse for TC1 cells and its derivatives) were injected in the right flank of vaccinated mice. Tumor growth was regularly monitored for the following weeks and the absence of tumors was considered as an indication of efficient antitumor vaccination.

To establish transplanted *s.c.* tumors, half a million wild type MCA205, TC1, as well as *Anxa1*$^{-/-}$ MCA205, *Hmgb1*$^{-/-}$ MCA205, *Anxa1*$^{-/-}$ TC1, *Hmgb1*$^{-/-}$ TC1 cells were *s.c.* inoculated into the right flank of wild type C57BL/6 mice or athymic *nu/nu* mice which were randomly assigned into treatment groups (n = 6–10 per group). When the tumor surface (calculated as longest dimension ×

perpendicular dimension × π/4) reached around 20–25 mm$^2$ (defined as day 0), mice received the treatments described below. Tumor surface was then monitored every 2–3 days and animals bearing neoplastic lesions that exceeded 250 mm$^2$ were euthanized.

To establish the orthotopic TC1 NSCLC model, wild type TC1 *Luc* cells ($5 \times 10^5$ in 100 μL PBS) were intravenously injected to wild type C57BL/6 mice or athymic nude (nu/nu) mice. Tumor incidence and development were monitored by in vivo photonic imaging of tumor cells' luciferase activity. About 7 days after injection, tumor incidence in the lung was detected at an exposure time of 4 min, and mice were assigned to different groups for treatment as described below.

To establish LLC1 orthotopically, C57BL/6 mice were anesthetized with 3% isoflurane and maintained in a position of dorsal decubitus. The fur at the area of surgery was removed to avoid contamination and 70% ethanol and iodide were used for sterilization. A lateral incision was made on the chest wall of each mouse and $2 \times 10^5$ LLC1 *Luc* cells in 50 μl PBS containing 20% matrigel matrix (Corning, Ref# 356231) were injected percutaneously into the left lung of the animals using 0.3-mL insulin syringes with 30 G hypodermic needles. The skin incisions were closed with a surgical skin clip. Tumor incidence and tumor growth were monitored by in vivo photonic imaging of tumor cells' luciferase activity. About 5 days after injection, tumor incidence in the lung was detected at an exposure time of 2 min, and mice were assigned to different treatment groups as described below.

For the acquisition of bioluminescence images, mice received an *i.p.* injection of luciferase substrate (Beetle Luciferin potassium salt, Promega) at a dose of 3 mg per mouse, and 8 min (for TC1 model), or 12 min (for the LLC1 model) post luciferin inoculation, photons were acquired on a Xenogen IVIS 50 bioluminescence in vivo imaging system (Caliper Life Sciences Inc., Hopkinton, MA, USA). In vivo imaging was conducted every 4–5 days with an exposure time starting with 4 min, which then gradually decreased to 3 min, 2 min, 1 min when photon saturation occurred. Tumor bearing mice showing photon saturation at 1 min of exposure were euthanized.

All treatments on mice bearing transplanted cancers are designed as following: Solvent for chemicals (Sol): 10% Tween-80, 10% PEG400, 4% DMSO, 76% physiological saline; v:v; *i.p.* mitoxantrone (MTX, 5.2 mg Kg$^{-1}$ in 200 μL Sol); *i.p.* cisplatin (CDDP, 0.25 mg Kg$^{-1}$ in 200 μL Sol); mitomycin C (MitoC, 0.25 mg Kg$^{-1}$ in 200 μL Sol); *i.t.* (R)/(S)-crizotinib (Criz, 25 mg Kg$^{-1}$ in 50 μL Sol); *i.p.* (R)/(S)-crizotinib (50 mg Kg$^{-1}$ in 200 μL Sol), at day 0 (when tumor became detectable) and day 2. In the case of consolidating with checkpoint blockade, mice also received *i.p.* injection of 200 μg anti-PD-1 antibody (Clone 29 F.1A12, BioXcell, West Lebanon, NH, USA), 100 μg anti-CTLA-4 antibody (Clone 9D9, BioXcell), alone or in combination, in 100 μL PBS or 300 μg isotype Ab (Clone LTF-2, BioXcell) in 100 μL PBS at day 8, 12 and 16; in the case of IFNγ neutralization, mice received an *i.p.* injection of 300 μg anti-IFNγ antibody (Clone R4-6A2, BioXcell) or equivalent isotype Ab (Clone HRPN, BioXcell) before chemotherapy, and were continuously administrated weekly; in the case of IFNAR neutralization, mice received an *i.t.* (for *s.c.* tumors) injection of 50 μg or *i.v.* (for orthotopic tumors) injection of 200 μg anti-IFNAR-1 antibody (Clone MAR1-5A3, BioXcell) or equivalent isotype antibody (Clone MOPC-21, BioXcell) before chemotherapy, and were continuously administrated at day 2, 4, 7 after the 1st chemotherapy; in the case of IL-12b neutralization, mice received an *i.p.* injection of 50 μg anti-IL-12 p40 antibody (Clone C17.8, BioXcell) or equivalent isotype antibody (Clone 2A3, BioXcell) before chemotherapy. Neutralization was continued by antibody administration every 5 days.

For the depletion of T cells, mice received an *i.p.* injection of 100 μg anti-CD8 (clone 2.43 BioXCell), anti-CD4 (clone GK1.5 BioXCell), or their combination at day −1 and day 0. Following the antibodies were supplied once a week for two weeks. For CD11b neutralization, mice received 100 μg anti-CD11b (clone M1/70 BioXCell) or equal amounts of isotype control antibody (clone LTF-2 BioXCell) at day −1, day 0. The injection was repeated every 2 days for the following 2 weeks.

**Adoptive plenocytes transfer**. The spleens from mice that had been cured from TC1 orthotopic lung cancers by the combination of (R)-crizotinib plus CDDP plus antiPD-1 antibody were excised, cut to small pieces and dissociated by passing through a 70 μm cell strainer. Then the cells were resuspended in erythrocyte lysis buffer. After centrifugation, cells from each spleen were resuspended in 250 μL PBS for *i.v.* injection (125 μl per mouse). One day after splenocyte transfer (SPT) mice were rechallenged with MCA205 cells ($1 \times 10^5$ per mouse) and TC1 luc cells ($2 \times 10^5$ per mouse) on the left and right flanks, respectively. Tumor growth was monitored regularly. Naïve mice injected with PBS only were used as negative controls, and cured mice were used as positive controls.

**Tissue section and immunofluorescence**. Subcutaneous tumors were established and mice were treated as described above (all drugs were administered through *i.p.* injection). Eight days after the first treatment tumor samples were harvested and rinsed briefly with ice cold PBS before cutting and fixation in 10% neutral buffered formalin (Sigma-Aldrich) for 4 h at room temperature. The tumors were then immersed in 30% sucrose (v:v in PBS, 50 mL per tumor), embedded in Tissue-Tek® O.C.T. Compound (Sakura, Villeneuve d'Ascq, France) and finally frozen at −80 °C. Five μm-thick serial tissue sections were generated on a Cryostat CM3050 S (Leica, Wetzlar, Germany) and captured on poly-L-lysine-coated slides (Fisher Scientific, Pittsburgh, PA, USA). For immunofluorescence staining of tissue

sections, slides were washed three times with TBS containing 0.05% Triton-X-100 (TBST) to remove remaining O.C.T. After blocking non-specific binding of antibodies with 10% fetal born serum (FBS) plus 1% bovine serum albumin (BSA) at room temperature for 1 h, samples were stained with specific primary antibodies in a humidified chamber (overnight at 4 ºC). If necessary, Alexa Fluor®-conjugated secondary antibodies (Molecular Probes-Life Technologies Inc.) were used at 1:200 dilution after washing three times for 5 min with TBST. After additional washing steps, samples were mounted with cover slides using Fluoroshield™ with DAPI (Sigma-Aldrich), which allowed nuclear counterstaining. For each sample, 10 view-fields from different sections of different layers were captured with a HR-SP8 Confocal Microscope (Leica). Image analysis was performed with the LAS X software (Leica).

**Urethane induced lung cancer model**. This protocol was approved by the Ethics Committee of Suzhou Institute of Systems Medicine, Chinese Academy of Medical Sciences (No: 2017AWEC011). Naïve FVB mice (female, aged between 6–7 weeks, weighing 20–22 g) were purchased from Charles River, Beijing, China. All mice received *i.p.* injections of urethane (dissolved in 0.9% NaCl saline solution, 1 g per Kg of body weight) once per week, for total of 6 weeks. Crizotinib was dissolved in solvent (10% PEG400, 10% TWEEN80, 4% DMSO, 76% NaCl saline solution, all percentages resemble volume/volume). CDDP was dissolved firstly in PBS, and then diluted with solvent before injection. On day 23, 25, 50, and 52 after the first injection of urethane, mice were treated with anticancer drugs (50 mg Kg$^{-1}$ crizotinib, 0.25 mg Kg$^{-1}$ CDDP or their combination) or the solvent control. On day 111-112, the presence of neoplastic nodules in mice were examined by whole body scanning with U-SPECT + /CT (MILabs, The Netherlands). The voxel resolution applied was 2.5 μm, and all mice were anesthetized with inhalational isoflurane during scanning. All mice were sacrificed on day 120–124 to harvest the lungs for immunohistochemical analysis.

Freshly harvested lung tissues were kept in 1× PBS on ice. A stereo microscope (Motic SMZ168TP) equipped with a high-resolution camera (Sony HD 1/1.8 inch color CCD) was used to record the images of all lung lobes (two sides). The number and sizes of all neoplastic nodules were quantified with the supporting software EZ-NET™.

All samples were then fixed in 4% PFA and embedded in O.C.T to be frozen at −80 °C for tissue sections. Frozen sections were adapted to room temperature and washed 3 times before hematoxylin-eosin staining using a hematoxylin-eosin staining kit, Sangon Biotech (Shanghai, China). Alternatively, samples were permeabilized with 0.5% Triton X100 and incubated with Alexa Fluor 488-conjugated anti-mouse IFN-γ antibody (clone XMG1.2, 5 μg ml$^{-1}$) before Alexa Fluor 647-conjugated anti-mouse CD3 antibody (clone 17A2, 5 μg ml$^{-1}$). Images were captured with a TCS SP8 confocal microscope (Leica). For each group, two nonconsecutive tissue sections were stained, and around 50 view fields were recorded for quantification.

**Kras$^{LSL−G12D/+}$;Trp53$^{flox/flox}$ (KP) lung cancer model**. KP mice (129 background) were bred in the Pittet Lab at the Massachusetts General Hospital, Harvard Medical School. Related animal experiments were approved by the Massachusetts General Hospital Subcommittee on Research Animal Care. KP NSCLC were induced by infecting KP mice intratracheally with an adenovirus expressing Cre recombinase (AdCre) that causes the activation of mutant Kras under the control of a Lox-Stop-Lox cassette and the inactivation of floxed Trp53, as described previously[31,46]. AdCre was obtained from the University of Iowa Gene Transfer Vector Core. Eight weeks post AdCre infection, the KP mice started to accept the treatments of 0.25 mg Kg$^{-1}$ *i.p.* CDDP in 200 μL solvent, 50 mg Kg$^{-1}$ *i.p.* (R)-crizotinib in 200 μL solvent, 0.25 mg Kg$^{-1}$ *i.p.* CDDP combined with 50 mg Kg$^{-1}$ *i.p.* (R)-crizotinib mixed in 200 μL solvent, or an equivalent volume of solvent (10% Tween-80, 10% PEG400, 4% DMSO, 76% physiological saline; v:v) once a week, repeating 3 weeks before the mice were euthanized for lung harvest. Evaluation of lung tumor burden was achieved by lung weight measurements, as well as histological analyses of lung tumor areas based on hematoxylin-eosin staining of explanted lung tissues. Hematoxylin-eosin and immunohistochemistry stainings were performed in the Hynes lab at the Massachusetts Institute of Technology as described previously[31]. Primary antibodies used for IHC were rat anti-mouse CD8a (clone 4SM15, eBioscience, SanDiego, CA, USA) and rabbit anti-Foxp3 (polyclonal, Novus Biologicals, Littelton, CO, USA). The NanoZoomer 2.0-RS slide scanner system (Hamamatsu) was used for image documentation and the Aperio Image-Scope Viewer (Leica Biosystems) was used for image quantification.

**In vitro T cell differentiation**. Naïve CD4$^+$ T cells (CD4$^+$CD62L$^{hi}$CD44$^{lo}$) were purified from spleen and lymph nodes. Isolated naïve CD4$^+$ T cells were stimulated with plate-bound antibodies against CD3 (clone 145-2C11, 2 μg mL$^{-1}$, BioXcell) and CD28 (clone PV-1, 2 μg mL$^{-1}$, BioXcell) and polarized into effector CD4$^+$ T lymphocyte subsets without cytokines (TH0 cells), or with IL-12 (20 ng mL$^{-1}$) for TH1 cells, or with IL-4 (20 ng mL$^{-1}$) for TH2 cells, or with TGF-β (2 ng mL$^{-1}$) for Treg cells, or with TGF-β (2 ng mL$^{-1}$) and IL-6 (25 ng mL$^{-1}$) for TH17 cells, or with TGF-β (2 ng mL$^{-1}$) and IL-4 (20 ng mL$^{-1}$) for TH9 cells. After 72 h of polarization in the presence of increasing doses of R-crizotinib, cell culture supernatants were assayed by ELISA for mouse IFN-γ (BD Biosciences), IL-17

(BioLegend, San Diego, CA, USA), and IL-9 (BioLegend) according to the manufacturer's protocol. Likewise, after three days, total RNA from T cells was extracted with TriReagent (Ambion, Austin, TX, USA), reverse transcribed using M-MLV Reverse Transcriptase (Invitrogen) and was analyzed by real-time quantitative PCR (RT-qPCR) with the Sybr Green method according to the manufacturer's instructions using the 7500 Fast Real-Time PCR system (Applied Biosystems). Expression was normalized to the expression of mouse Actb. Primers designed to assess gene expression are described in Supplementary Table 1.

**Isolation and phenotyping of local lymphocytes.** Tumor-free C57BL/6 mice were randomly assigned into different groups ($n = 5$ per group) and subjected to the following treatments similarly to mice with established cancers (see above): Vehicle only ("solvent"), CDDP only, crizotinib only or combined treatment with CDDP and Crizotinib. After 8 days, splenic cells were harvested and re-stimulated with phorbol 12-myristate 13-acetate (PMA) and ionomycin and cells were analyzed using flow cytometry as described below.

**Phenotyping of T cell exhaustion markers.** Blood was collected from tumor bearing mice and peripheral blood mononuclear cells (PBMCs) were isolated by Ficoll-Paque PLUS (GE Healthcare) density gradient centrifugation. PBMCs were incubated with LIVE/DEAD® dye (Invitrogen) and incubated with antibodies specific for CD16/CD32 for Fc block as described below, and subjected to surface staining with fluorescent antibodies specific for CD3e, CD4, CD8, CTLA-4, LAG-3, PD-1, and TIM-3 (detailed information are provided in the following antibody list).

**Isolation and phenotyping of tumor-infiltrating lymphocytes.** Subcutaneous MCA205 or TC1 cancers were established and tumor bearing mice were treated as described above. Animals were sacrificed and tumors were excised and immediately collected in gentleMACS C Tubes (Miltenyi Biotech; Bergisch Gladbach, Germany) containing ≥ 1 mL RPMI-1640 medium. The samples were kept on ice until dissociation using the gentleMACS dissociator with a Miltenyi mouse tumor dissociation kit (Miltenyi Biotech) according to the manufacturer's protocol. The dissociated bulk tumor cell suspension was resuspended in RPMI-1640, sequentially passed through 70 μm and 30 μm nylon cell strainers (Miltenyi Biotec) and washed twice with cold PBS. To test IFNα and IL-17 production, cells were restimulated with ionomycin (1 μg mL$^{-1}$, Sigma-Aldrich) plus PMA (50 ng mL$^{-1}$, Sigma-Aldrich) in the presence of BD GolgiPlug (1:1000, BD Bioscience) in CTL-Test™ Medium (Cellular Technology Limited, Cleveland, OH, USA) containing 2 mM L-Glutamine for 6 h before the following steps. Prior to surface staining of fluorescent antibodies, samples were incubated with LIVE/DEAD® Fixable Yellow Dead Cell dye (Invitrogen) to discriminate viable cells from damaged/dead cells, and incubated with antibodies against CD16/CD32 (clone 2.4G2, BD Biosciences) to block Fc receptors. Finally, cells were incubated with a panel of fluorescence-conjugated antibodies (surface staining) before permeabilization and fixation with Foxp3/Transcription Factor Staining Buffer Set (eBioscience, San Diego,CA, USA) for the staining of intracellular Foxp3, IFNγ, and IL-17. Data were acquired on a BD LSRFortessa flow cytometer (BD Biosciences) and analyzed by means of FlowJo software. Absolute counts of tumor-infiltrating lymphocytes were obtained taking in account the weight of the harvested tumor, total volume of the dissociated tumor cell suspension (cell concentration typically set to 250 mg ml$^{-1}$ in PBS), proportion of the whole cell suspension stained (typically 200 μl containing 50 mg of bulk tumor cell suspension) and proportion of the stained cell suspension ran through the flow cytometer (typically ~300 out of 400 μl of the stained cell suspension). Detailed gating strategy for the flow cytometric analysis can be found in corresponding Supplementary Figures.

**Antibodies for immunofluorescence and flow cytometry analysis.** Anti-CD3 antibody (ab5690) and Anti-CD11c antibody (clone 3.9, ab11029) were purchased from Abcam (Cambridge, UK). Alexa Fluor® 488-conjugated anti-mouse CD86 antibody (clone GL-1), Alexa Fluor® 488 or Alexa Fluor® 594-conjugated anti-mouse CD8a antibody (clone 53-6.7), Pacific blue anti-mouse CD8a (Clone 53-6.7), APC anti-mouse CD3ε (Clone 145-2C1), Alexa Fluor® 488-conjugated anti-mouse Foxp3 antibody (clone MF-14), and PE-Cy7-conjugated anti-mouse CD152/CTLA-4 antibody (clone UC10-4B9) were purchased from BioLegend. BV421-conjugated anti-mouse CD278/ICOS (clone 7E.17G9), FITC or PE-conjugated anti-mouse CD8a (clone 53-6.7), FITC anti-mouse CD4 (RM4-4), PE and APC anti-mouse IFNγ (XMG1.2), PE Rat anti-mouse IL4 (1B11), BV605 anti-mouse IL17A (TC11-18H10), PE-Cy7-conjugated anti-mouse CD11c (clone HL3) were purchased from BD Bioscience (San José, CA, USA). PerCP-Cy5.5 or pacific blue-conjugated anti-mouse CD4 (clone RM4-5), PE-Cy7-conjugated anti-mouse CD25 (clone PC61 5.3), Foxp3 efluor450 (clone FJS-16s), APC or FITC-conjugated anti-mouse CD3g,d,e (clone 17 A2), APC-eFluor780, APC-Cy7-conjugated anti-mouse CD279/PD-1 (clone J43), PE-conjugated anti-mouse IL-17A (clone eBio17B7), Percp_Cy5.5-conjugated anti-mouse CD3e (clone 145-2C11), FITC-conjugated anti-mouse LAG-3 (clone eBioC9B7W), PE-conjugated anti-mouse PD-1 (clone J43), APC-conjugated anti-mouse TIM-3 (clone 8B.2C12), APC-conjugated anti-mouse TIM-3 (clone 8B.2C12), pacific blue-conjugated anti-mouse CD11b (clone M1/70.15), Alexa Fluor® 488-conjugated anti-mouse Ly-6C

(clone HK1.4), and PE-conjugated anti-mouse Ly-6G (clone 1A8-Ly6g) were purchased from eBioscience (San Diego, CA, USA).

**RNA-Seq (RNA sequencing) analyses of chemotherapy impacted gene expression.** To prepare RNA samples for RNA-sequencing analysis, s.c. tumors were established by injecting $5 \times 10^5$ WT MCA205 cells (in 100 μL PBS) into the right flank of WT C57BL/6 mice. When tumors became palpable (approximately 7 days post cell injection), mice received 0.25 mg Kg$^{-1}$ i.p. CDDP in 200 μL solvent, 50 mg Kg$^{-1}$ i.p. (R)-crizotinib in 200 μL solvent, 0.25 mg Kg$^{-1}$ i.p. CDDP dissolved together with 50 mg Kg$^{-1}$ i.p. (R)-crizotinib in 200 μL solvent, or an equivalent volume of solvent at day 0 and day 2. At day 10 tumors were harvested and immediately immersed in the RNAlater RNA Stabilization Reagent (Qiagen, Hilden, Germany) and total RNA was extracted with the RNeasy Plus Mini Kit (Qiagen) following the manufacturers' instructions.

RNA-Seq data analysis was performed by GenoSplice technology (www.genosplice.com). Sequencing, data quality, reads repartition (e.g., for potential ribosomal contamination), and insert size estimation were performed using FastQC, Picard-Tools, Samtools, and rseqc. Reads were mapped using STARv2.4.0[47] on the mm10 Mouse genome assembly. Gene expression regulation study was performed as already described[48]. Briefly, for each gene present in the Mouse FAST DB v2016_1 annotations, reads aligning on constitutive regions (that are not prone to alternative splicing) were counted. Based on these read counts, normalization and differential gene expression were performed using DESeq2[49] on R (v.3.2.5). Only genes expressed in at least one of the two compared experimental conditions were further analyzed. Genes were considered as expressed if their rpkm value was greater than 95% of the background rpkm value based on intergenic regions. Results were considered statistically significant for corrected $p$-values ≤ 0.05 and fold-changes ≥ 1.2.

Analysis for enriched GO terms, KEGG pathways and REACTOME pathways were performed using DAVID Functional annotation Tool (v6.8). GO terms and pathways were considered as enriched if fold enrichment ≥ 2.0, uncorrected $p$-value ≤ 0.05 and minimum number of regulated genes in pathway/term ≥ 2.0. Analysis was performed three times: using all regulated genes, using upregulated genes and using downregulated genes only. Union of these three analyses was made to provide a single list of results.

**Assessment of hepatotoxicity.** MCA205 cells were s.c. inoculated into the right flank of C57BL/6 mice. When tumor became palpable mice received (R)-crizotinib alone (50 mg Kg$^{-1}$ i.p.) or in combination with CDDP (0.25 mg Kg$^{-1}$ i.p.), together with anti-PD-1 antibody (10 mg Kg$^{-1}$ i.p.) in a simultaneous (schedule A) or sequential (schedule B) way. Schedule A consisted of (R)-crizotinib and CDDP administration combined with anti-PD-1 at day 0, 2, and 4 followed by (R)-crizotinib and CDDP at day 6 to simulate continuous crizotinib administration. Schedule B consisted of (R)-crizotinib and CDDP administration at day 0 and 2 followed by anti-PD-1 at day 8, 12, 16. The mice were observed daily and body-weight was recorded as one proxy for toxicity. Blood and liver were sampled 3 days after the last antibody administration. Serum alanine transaminase activity (ALT) was measured using the Alanine Transaminase Activity Assay Kit (Colorimetric/Fluorometric) (Abcam, ab105134) following the manufacture's protocol. Livers were fixed in 4% PFA and subjected to hematoxylin and eosin staining for morphology observation.

**Statistical analyses.** Unless specified, results were expressed as mean ± s.e.m. Statistical tests included unpaired one-tailed and two-tailed Student's $t$-tests using Welch's correction and one-way ANOVA followed by multiple comparison tests. When applicable, two-way ANOVA followed by multiple comparison tests were used for analysis of interleaved box whiskers plot and scatter dots plots; comparison of tumor growth curves was performed by means of the ANOVA Type 2 (Wald test); Comparison of survival curves was performed with the Likelihood ratio test. $P$-values of 0.05 or less were considered to denote significance (*$p < 0.05$; **$p < 0.01$; ***$p < 0.001$; ns, not significant). In vivo data was analyzed using the freely available TumGrowth software (https://github.com/kroemerlab/TumGrowth)[50].

**Reporting Summary.** Further information on experimental design is available in the Nature Research Reporting Summary linked to this article.

## Data availability
The RNASeq data have been deposited in the NCBI GEO database under the accession code GSE126988, The source data underlying Figs. 1 and 6 are provided as a Source Data file. All the other data supporting the findings of this study are available within the article and its supplementary information files and from the corresponding author upon reasonable request. A reporting summary for this article is available as a Supplementary Information file.

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

## Acknowledgements

G.K. is supported by the Ligue contre le Cancer Comité de Charente-Maritime (équipe labelisée); Agence National de la Recherche (ANR)–Projets blancs; ANR under the frame of E-Rare-2, the ERA-Net for Research on Rare Diseases; Association pour la recherche sur le cancer (ARC); Cancéropôle Ile-de-France; Chancelerie des universités de Paris (Legs Poix), Fondation pour la Recherche Médicale (FRM); a donation by Elior; the European Commission (ArtForce); the European Research Council (ERC); Fondation Carrefour; Institut National du Cancer (INCa); Inserm (HTE); Institut Universitaire de France; LeDucq Foundation; the LabEx Immuno-Oncology; the RHU Torino Lumière; the Seerave Foundation; the SIRIC Stratified Oncology Cell DNA Repair and Tumor Immune Elimination (SOCRATE); the SIRIC Cancer Research and Personalized Medicine (CARPEM); and the Paris Alliance of Cancer Research Institutes (PACRI). L.Z. is supported by the Ligue contre le Cancer Comité de Charente-Maritime. P.L. is supported by the China Scholarship Council. A.P., V.M.-C. and G.K. were supported by Horizon 2020/European Union (Alkatraz). C.P. was supported by a MGH ECOR Tosteson Postdoctoral Fellowship. S.R. was supported by a Metastasis/Cancer Research Postdoc fellowship from the MIT Ludwig Center for Molecular Oncology Research and funding from Richard O. Hynes (NIH grant U54-CA163109 and Howard Hughes Medical Institute). Y.M. is supported by Natural Science Foundation of China (NSFC, Grant No 81722037 and 81671630), China Ministry of Science and Technology (National key research and development program, Grant No 2017YFA0506200), Natural Science Foundation of Jiangsu Province (Grant No BK20170006 and BK20160379), Chinese National Thousand Talents Program, and CAMS Initiative for Innovative Medicine (CAMS-I2M, 2016-I2M-1-005). We thank the members of the Hope Babette Tang Histology Facility at the Koch Institute Swanson Biotechnology Center for technical support. L.A. is supported by the ERC (grant No 677251).

## Author contributions

P.L., L.Z., J.P., S.L., Ad.P., T.Y., K.I., L.B., E.V. performed most of the experiments; V.S. performed the Seahorse analysis; C.P. and C.E. performed the in vivo experiments with the KP lung cancer model; S.R. performed the immunohistochemistry of KP mice samples; A.M. and T.M. tested the immune effects of crizotinib on the mouse immune

system, X.L., H.Y., Q.L., J.C. performed the in vivo experiments with urethane induced lung cancer model; L.S. performed the intercostal injection for the establishment of LLC1 orthotopic lung cancer model; S.D., F.A., D.L., S.B.An.P. and A.B. measured drug concentrations; O.K. and G.K. conceived the study; V.M.-C., E.T., L.Z., L.A., Y.M., M.J.P., designed (parts of) the study; P.L., L.Z., O.K., and G.K. wrote the paper.

## Additional information

**Competing interests:** The authors declare no competing interests.

