## [Peer Review File · Nature Communications]

Reviewers' Comments:

Reviewer #1:

Remarks to the Author:

REF. NCOMMS-18-25339 by Kroemer

The paper by the group of Kroemer And Kepp (Liu et al.) provides a major extension of the concept that chemotherapeutic agents capable of inducing immunogenic cell death (ICD) (such as anthracyclines and oxaliplatin) can stimulate anticancer immune responses that are therapeutically relevant, meaning that they contribute to tumor growth reduction. Indeed, the group headed by the PI has pioneered the concept that ICD might have a major clinical impact, explaining the success of (some) anticancer chemotherapies in immunological terms. Here, the authors show that the tyrosine kinase inhibitor, crizotinib, if used at high doses, can stimulate ICD, presumably via an off-target effect (most likely involving the combined inhibition of several tyrosine kinase), if combined with agents that usually fail to induce ICD (such as cisplatin or mitomycin C). The authors provide an exhaustive preclinical and mechanistic characterization of the interaction between crizotinib and cisplatin to conclude that both agents synergize to cause tumor growth reduction in multiple lung cancer models via the stimulation of T lymphocyte-mediated anticancer immune responses. Moreover, the authors show that the combination of crizotinib and cisplatin can be used to sensitize orthotopic lung cancers to immunotherapy with PD-1 blockade, in suitable preclinical models.

The data are innovative and of high quality and would merit the the interest of scientific community, offering innovative potential therapeutic venues. Altogether, the results are convincing and well presented, the authors may consider ameliorating or expanding their paper in the following points:

-1- Given the cardinal importance of IL-12 for anticancer immune responses, is there an effect of crizotinib (alone or in combination with cisplatin) on IL-12 production by dendritic cells?

-2- Supplementary figure e should report also the data on the oxugen consumbtion rate. In fact ECAR would better integrated also with OCR.

-3- The authors focus the analysis of crizozitinib effect on the T cell infiltrate within the tumor bed. What about other lymphoid elements such as NK cells?

-4- Does crizotinib affect tumor infiltration by myeloid cells (such as macrophages and MDSC)? It may be important to obtain a more complete picture of the effects of crizotinib on the tumor immune infiltrate.

-5- The authors show that crizotinib induces PD-L1 expression by a restricted panel of non-small cell lung cancer cell lines. What about other cancer types? The answer to this question might open the door to a more extended use of crizotinib for immunotherapy sensitization in a large panel of organ-specific tumors.

-6- The authors should discuss the putative mechanisms why co-treatment of crizotinib and PD-1 blockade is hepatotoxic, while sequential treatment (first crizotinib plus chemotherapy, then PD-1 blockade) is not.

-7- The manuscript would benefit of adding a supplementary figure with a graphic abstract of the procedure for the unbiased screen.

Reviewer #2:

Remarks to the Author:

The manuscript from Liu et al initially presents results from a fluorescence-based in vitro screen with a library of tyrosine kinase inhibitors to identify those which may induce an immunogenic cell death. They identified crizotinib as a top lead hit and provide strong in vitro data to confirm it as such. When combined with chemotherapeutic agents such as cisplatin and mitomycin C, crizotinib causes an anti-neoplastic effect, which is dependent on functioning T lymphocytes and IFN γ signaling. Using multiple in vivo models of lung cancer, the authors show that this combination increases T cell infiltration and IFN γ signaling. Additionally, tumors treated with cisplatin with crizotinib have increased expression of immunosuppressive markers PD-1, CTLA-4, and PD-L1, and the addition of immune checkpoint inhibitors after treatment with crizotinib and cisplatin effectively eliminates tumors. The work is performed well, with appropriate controls and the manuscript is clearly written and easy to follow. The manuscript provides strong preclinical evidence for the efficacy of the combination of crizotinib with chemotherapy and immune checkpoint inhibition in the treatment of NSCLC. While providing convincing evidence that the combination of crizotinib, cisplatin, and immune checkpoint inhibitors is efficacious in preclinical models of lung cancer, the manuscript could improve upon the suggested mechanism for the interaction between cisplatin and crizotinib in inhibiting tumor progression. The immune data provided (i.e. immune infiltrate and IFN γ signaling) does not sufficiently address why the combination effectively controls tumor growth while single agent crizotinib does not; thus, this should be more thoroughly investigated. If the authors address this concern along with the other suggestions and comments outlined below, this manuscript would fit well within the scope of Nature Communications.

Detailed Comments:

1. The authors should provide a clearer justification of why they chose to take R-crizotinib forward as the best hit from the in vitro screen when multiple other TKIs (e.g., foretinib, canertinib, lestaurtinib) also induced ICD.
2. It is not initially apparent why the combination with cisplatin is necessary to inhibit tumor progression compared to crizotinib alone. All the ICD-related phenotypes (with the exception of calreticulin redistribution), MHC upregulation, the CD8 $^+$ T cell and CD11c $^+$ /CD86 $^+$ dendritic cell infiltration, and the IFN γ response are all upregulated by crizotinib alone. Certainly the in vitro ICD screen was drug alone. Why is there not an effect of crizotinib alone on tumor growth (except in the Kras/p53 GEMM which does show a marginal response to single agent crizotinib)? What is the mechanism of synergy between CDDP and crizotinib to mediate tumor cell killing? The immune data presented is not a convincing argument since single agent crizotinib does have an impact on each of the assayed immune populations.
3. Figure 2: Were any non-crizotinib target genes used in the siRNA knockdown studies to determine whether the effect was specific on these particular targets? Similarly, are the crizotinib targets (JAK2, ALK, etc) downregulated in tumors that are treated with crizotinib? It would be good to show which targeted are inhibited in vivo.
4. The data showing that the impact of CDDP and crizotinib in nude mice no longer has an antineoplastic effect is convincing (Fig S4). However, this could be strengthened by performing CD8 or CD4 depletion studies in immunocompetent mice to further narrow down the important subpopulation(s) in the immune competent context.
5. Was depletion of antigen presenting cells tested to determine the role of the CD11c/CD86 dendritic cells in the crizotinib + CDDP mediated cell killing?
6. CD8 T cell exhaustion is not explored (only total tumor RNA levels and CD4 exhaustion – Fig 6). Flow cytometry using exhaustive markers like TIM3, LAG3, and PD1 should be completed to analyze the extent of CD8 exhaustion and supplement the reasoning for the addition of immune checkpoint inhibitors.
7. Splenocyte or tumor-infiltrating lymphocyte transfer could be used in these studies. Taking splenocytes from cured animals and implanting into naive mice prior to tumor cell challenge would nicely demonstrate an immune-related protection.
8. The combination of immune checkpoint inhibitors with crizotinib + CDDP should be shown in another model besides the orthotopic and s.c. models. Either the GEMM or the urethane-induced model would be a good addition here.
9. (S)-crizotinib seems to combine with cisplatin to mediate tumor cell killing in vivo (Fig S4b),

despite not being as strong of an ICD inducer as (R)-criz. It is possible that this combination then may not be dependent upon whether the drug induces ICD? This is related to point 1, wherein a mechanism of synergy between cisplatin and crizotinib needs to be further elucidated.

10. Did you combine (S)-crizotinib with CDDP in immunodeficient mice? Additionally, have you done this combination along with treatment with immune checkpoint inhibitors? Again, this would indicate whether ICD is relevant.

1. The translational potential of this work is unclear and somewhat confusing. As outlined in the text, the effects of crizotinib or other ALK/ROS TKIs in patients appear to be dependent on on-target effects, without an apparent immune basis to the clinical responses. Why would crizotinib not induce ICD and an enhanced immune response in patient tumors? Is it due to other effects, for example in patients treated with crizotinib, do immunosuppressive markers such as PD-1/PD-L1 become upregulated?

Conversely, the authors provide no data that the effect of ICD is cancer cell-specific, raising the possibility that the observed effects will also occur in normal cells and confer significant immune-mediated to the combination treatments. In fact, clinically, in both concurrent and sequential treatment with ALK inhibitors and PD-(L)1 inhibitors, significant immune-related toxicity has been described. Might this mechanism help to explain the clinical observations? This should at least be addressed more completely in the Discussion, but the authors may have analyses of normal tissues from the experiments that will shed light on this.

2. Typos: line 158- fibrosarcoma; line 168- others

Point-by-point reply to reviewer #1

General remarks by the referee: The paper by the group of Kroemer and Kepp (Liu et al.) provides a major extension of the concept that chemotherapeutic agents capable of inducing immunogenic cell death (ICD) (such as anthracyclines and oxaliplatin) can stimulate anticancer immune responses that are therapeutically relevant, meaning that they contribute to tumor growth reduction. Indeed, the group headed by the PI has pioneered the concept that ICD might have a major clinical impact, explaining the success of (some) anticancer chemotherapies in immunological terms. Here, the authors show that the tyrosine kinase inhibitor, crizotinib, if used at high doses, can stimulate ICD, presumably via an off-target effect (most likely involving the combined inhibition of several tyrosine kinase), if combined with agents that usually fail to induce ICD (such as cisplatin or mitomycin C). The authors provide an exhaustive preclinical and mechanistic characterization of the interaction between crizotinib and cisplatin to conclude that both agents synergize to cause tumor growth reduction in multiple lung cancer models via the stimulation of T lymphocyte-mediated anticancer immune responses. Moreover, the authors show that the combination of crizotinib and cisplatin can be used to sensitize orthotopic lung cancers to immunotherapy with PD-1 blockade, in suitable preclinical models.

The data are innovative and of high quality and would merit the interest of the scientific community, offering innovative potential therapeutic venues. Altogether, the results are convincing and well presented.

Our response: We thank the reviewer for the positive comments and the encouragement. We have responded to each of her/his comments, as detailed below.

Suggestion No. 1 by reviewer 1: Given the cardinal importance of IL-12 for anticancer immune responses, is there an effect of crizotinib (alone or in combination with cisplatin) on IL-12 production by dendritic cells?

Our response: We measured the effect of crizotinib on IL-12 production by bone marrow-derived dendritic cells (BM-DC), finding that crizotinib indeed induces a small but significant increase in IL-12 production. Indeed, the stimulation of IL-12 production by crizotinib was much smaller than that obtained with the positive control, bacterial lipopolysaccharide (LPS). This information has been added to supplementary Fig. 12g-i.

Suggestion No. 2 by reviewer 1: Supplementary figure e should report also the data on the oxygen consumption rate. In fact, ECAR would better integrated also with OCR.

Our response: We measured OCR in addition to ECAR, and we added these data to the supplementary Fig. 2e,f.

Suggestion No. 3 by reviewer 1: The authors focus the analysis of crizotinib effect on the T cell infiltrate within the tumor bed. What about other lymphoid elements such as NK cells?

Suggestion No. 4 by reviewer 1: Does crizotinib affect tumor infiltration by myeloid cells (such as macrophages and MDSC)? It may be important to obtain a more complete picture of the effects of crizotinib on the tumor immune infiltrate.

Our response: We opted for responding to suggestions No. 3 and 4 at the same time, because both suggestions deal with the characterization of the immune infiltrate after crizotinib treatment. Multiparametric cytofluorometric analysis of the leukocytes present in the tumor bed upon treatment with crizotinib, alone or together with cisplatin, has been performed, revealing that crizotinib induces a significant increase in the infiltration of cancers by inflammatory macrophages and NK1.1⁺ $\gamma\delta$ T cells (Natural killer T, NKT cells) but no change in the total population of activated NK cells. In contrast, crizotinib does not affect the frequency of myeloid-derived suppressor cells (MDSC) in the tumor bed. These new results have been added to the paper, in supplementary Fig. 9 g-k.

Suggestion No. 5 by reviewer 1: The authors show that crizotinib induces PD-L1 expression by a restricted panel of non-small cell lung cancer cell lines. What about other cancer types? The answer to this question might open the door to a more extended use of crizotinib for immunotherapy sensitization in a large panel of organ-specific tumors.

Our response: We did the experiment suggested by the reviewer and found that crizotinib stimulates high expression of PD-L1 in different human/mouse cancer cell lines in a time/dose-dependent way. This includes colorectal cancer cell lines (human HCT116 and murine CT26). We have added these new data to the revised version of the paper, in supplementary Fig. 14.

Suggestion No. 6 by reviewer 1: The authors should discuss the putative mechanisms why co-treatment of crizotinib and PD-1 blockade is hepatotoxic, while sequential treatment (first crizotinib plus chemotherapy, then PD-1 blockade) is not.

Our response: We have added a short discussion on the possible mechanisms of toxic effects of simultaneous versus sequential administration of crizotinib and PD-1 blockade to the Discussion of the paper.

Suggestion No. 7 by reviewer 1: The manuscript would benefit of adding a supplementary figure with a graphic abstract of the procedure for the unbiased screen.

Our response: A graphic abstract has been added to the paper, reported as supplementary Fig. 19

NCOMMS-18-25339
Fist Author: Peng Liu
Last Author: Guido Kroemer
Revision date: Dec. 14

Point-by-point reply to reviewer #2

General remarks by the referee: The manuscript from Liu et al initially presents results from a fluorescence-based in vitro screen with a library of tyrosine kinase inhibitors to identify those which may induce an immunogenic cell death. They identified crizotinib as a top lead hit and provide strong in vitro data to confirm it as such. When combined with chemotherapeutic agents such as cisplatin and mitomycin C, crizotinib causes an anti-neoplastic effect, which is dependent on functioning T lymphocytes and IFN γ signaling. Using multiple in vivo models of lung cancer, the authors show that this combination increases T cell infiltration and IFN γ signaling. Additionally, tumors treated with cisplatin with crizotinib have increased expression of immunosuppressive markers PD-1, CTLA-4, and PD-L1, and the addition of immune checkpoint inhibitors after treatment with crizotinib and cisplatin effectively eliminates tumors. The work is performed well, with appropriate controls and the manuscript is clearly written and easy to follow. The manuscript provides strong preclinical evidence for the efficacy of the combination of crizotinib with chemotherapy and immune checkpoint inhibition in the treatment of NSCLC. While providing convincing evidence that the combination of crizotinib, cisplatin, and immune checkpoint inhibitors is efficacious in preclinical models of lung cancer, the manuscript could improve upon the suggested mechanism for the interaction between cisplatin and crizotinib in inhibiting tumor progression. The immune data provided (i.e. immune infiltrate and IFN γ signaling) does not sufficiently address why the combination effectively controls tumor growth while single agent crizotinib does not; thus, this should be more thoroughly investigated. If the authors address this concern along with the other suggestions and comments outlined below, this manuscript would fit well within the scope of Nature Communications.

Our response: We thank the reviewer for accurately summarizing our work and for the positive comments. The referee wants us to address more insights why the combination of crizotinib plus chemotherapy is more efficient than either crizotinib or chemotherapy alone. We have addressed this issue, as indicated below, in our response to the detailed comments of the reviewer.

Detailed comment No. 1 by reviewer #2: The authors should provide a clearer justification of why they chose to take R-crizotinib forward as the best hit from the in vitro screen when multiple other TKIs (e.g., foretinib, canertinib, lestaurtinib) also induced ICD.

Our response: We have added a phrase to the text to indicate why we have chosen to follow up by working with R-crizotinib rather than with other TKIs. Indeed, R-crizotinib ranked highest among FDA/EMA-approved drugs in the aggregate analysis of the two screens that we performed. Moreover, the clinical characterization of R-crizotinib has been more profound than that of the other non-approved TKIs (e.g., foretinib, canertinib, lestaurtinib) mentioned here.

Detailed comment No. 2 by reviewer #2: It is not initially apparent why the combination with cisplatin is necessary to inhibit tumor progression compared to crizotinib alone. All the ICD-related phenotypes (with the exception of calreticulin redistribution), MHC upregulation, the CD8⁺ T cell and CD11c⁺/CD86⁺ dendritic cell infiltration, and the IFN γ response are all upregulated by crizotinib alone. Certainly the *in vitro* ICD screen was drug alone. Why is there not an effect of crizotinib alone on tumor growth (except in the Kras/p53 GEMM which does show a marginal response to single agent crizotinib)? What is the mechanism of synergy between CDDP and crizotinib to mediate tumor cell killing? The immune data presented is not a convincing argument since single agent crizotinib does have an impact on each of the assayed immune populations.

Our response: We have performed clonogenic assays to show that CDDP efficiently abolishes clonogenicity, while crizotinib reduces clonogenicity but fails to eradicate the cancer cells, the combination is fully efficient in killing/arresting cancer cells *in vitro*. These results have been added as new supplementary Fig. 4. We believe that this is the reason why cancer cells treated with crizotinib alone form tumors when they are injected subcutaneously (which is not the case for cancer cells treated with cisplatin, alone or in combination with crizotinib: no tumors are formed). These results have been mentioned in the Results. For this reason, it is not possible to vaccinate mice with crizotinib-only-treated cells, hence explaining why we characterized the combination effect.

Detailed comment No. 3 by reviewer #2: Figure 2: Were any non-crizotinib target genes used in the siRNA knockdown studies to determine whether the effect was specific on these particular targets? Similarly, are the crizotinib targets (JAK2, ALK, etc) downregulated in tumors that are treated with crizotinib? It would be good to show which targeted are inhibited *in vivo*.

Our response: We knocked down a few other target genes (such as BTK, EGFR, ERBB, HCK) with validated siRNAs and found that these manipulations induced less CALR exposure, ATP release and HMGB release than treatment with crizotinib (new supplementary Fig. 3e-g). As suggested by the reviewer, we treated cancer cells with crizotinib *in vitro* and then measured the expression of potential target kinases without finding a significant decrease in the mRNAs of ALK, JAK2, MET, and ROS1. Similarly, MCA205 tumors treated with crizotinib *in vivo* failed to upregulate or downregulate Alk, Jak2, Met, and Ros1. These results have been added as supplementary Fig. 3l-n. They reveal that there is no major effect of crizotinib on the expression of these kinases.

Detailed comment No. 4 by reviewer #2: The data showing that the impact of CDDP and crizotinib in nude mice no longer has an antineoplastic effect is convincing (Fig S4). However, this could be strengthened by performing CD8 or CD4 depletion studies in immunocompetent mice to further narrow down the important subpopulation(s) in the immune competent context.

Our response: We performed the depletion of CD8⁺ cells alone, CD4⁺ cells alone or both together to show which T lymphocyte subpopulation is required for the combined CDDP/cisplatin effect. These results have been added as supplementary Fig. 7c-e.

Detailed comment No. 5 by reviewer #2: Was depletion of antigen presenting cells tested to determine the role of the CD11c/CD86 dendritic cells in the crizotinib + CDDP mediated cell killing?

Our response: We blocked CD11b to inhibit the extravasation of myeloid cells, showing that they are important for the effect of crizotinib + CDDP. These results have been added as supplementary Fig. 7f.

Detailed comment No. 6 by reviewer #2: CD8⁺ T cell exhaustion is not explored (only total tumor RNA levels and CD4 exhaustion – Fig 6). Flow cytometry using exhaustive markers like TIM3, LAG3, and PD1 should be completed to analyze the extent of CD8 exhaustion and supplement the reasoning for the addition of immune checkpoint inhibitors.

Our response: We performed cytofluorometric experiments for the detection of exhaustion markers. We found that LAG-3 and PD-1 (but not CTLA-4 nor TIM-3) were upregulated on circulating CD4 and CD8 T cells. These results have been added as supplementary Fig. 15

Detailed comment No. 7 by reviewer #2: Splenocyte or tumor-infiltrating lymphocyte transfer could be used in these studies. Taking splenocytes from cured animals and implanting into naïve mice prior to tumor cell challenge would nicely demonstrate an immune-related protection.

Our response: According to the referee's suggestion, we transferred splenocytes from cured animals to naïve mice before rechallenge. At difference to cured mice (which did not develop tumors upon rechallenge with TC1 cells), naïve mice that had received adoptively transferred splenocytes from cured mice, developed tumors shortly after rechallenge. However, these tumors were then spontaneously eliminated, indicating that the adoptive transfer of splenocytes from cured mice conferred immunity against the cancer cells. These results have been added as Supplementary Fig. 17 k-n

Detailed comment No. 8 by reviewer #2: The combination of immune checkpoint inhibitors with crizotinib + CDDP should be shown in another model besides the orthotopic and s.c. models. Either the GEMM or the urethane-induced model would be a good addition here.

Our response: The editors of Nature Communication imposed a three-month deadline for the revision of our manuscript. Unfortunately, the breeding of the GEMM or the preparation of the urethane-induced cancers would trespass this temporary threshold. Instead, we opted for showing the efficacy of the combination therapy in yet another orthotopic model, namely LLC1 tumors forming after intrathoracic injection of LLC1 cells. The combination of (R)-crizotinib, CDDP and PD-1 blockade achieved a 100% cure rate (15 out of 15 mice) against this kind of cancer. These results have been added as supplementary Fig. 17g,h.

Detailed comment No. 9 by reviewer #2: (S)-crizotinib seems to combine with cisplatin to mediate tumor cell killing in vivo (Fig S4b), despite not being as strong of an ICD inducer as (R)-criz. It is possible that this combination then may not be dependent upon whether the drug induces ICD? This is related to point 1, wherein a mechanism of synergy between cisplatin and crizotinib needs to be further elucidated.

Our response: We explored the efficacy of (S)-crizotinib as a cell death inducer in clonogenic assays and found that it was even less efficient than (R)-crizotinib in reducing clonogenicity. These results have been added as new supplementary Fig. 4

Detailed comment No. 10 by reviewer #2: Did you combine (S)-crizotinib with CDDP in immunodeficient mice? Additionally, have you done this combination along with treatment with immune checkpoint inhibitors? Again, this would indicate whether ICD is relevant.

Our response: In response to the reviewer's question, we tested whether this combination sensitized to immune checkpoint blockade. These negative results have been added to the paper in supplementary Fig. 17i,j.

Detailed comment No. 11 by reviewer #2: The translational potential of this work is unclear and somewhat confusing. As outlined in the text, the effects of crizotinib or other ALK/ROS TKIs in patients appear to be dependent on on-target effects, without an apparent immune basis to the clinical responses. Why would crizotinib not induce ICD and an enhanced immune response in patient tumors? Is it due to other effects, for example in patients treated with crizotinib, do immunosuppressive markers such as PD-1/PD-L1 become upregulated? Conversely, the authors provide no data that the effect of ICD is cancer cell-specific, raising the possibility that the observed effects will also occur in normal cells and confer significant immune-mediated to the combination treatments. In fact, clinically, in both concurrent and sequential treatment with ALK inhibitors and PD-(L)1 inhibitors, significant immune-related toxicity has been described. Might this mechanism help to explain the clinical observations? This should at least be addressed more completely in the Discussion, but the authors may have analyses of normal tissues from the experiments that will shed light on this.

Our response: The reviewer raises an excellent point. Driven by her/his comments, we determined the capacity of crizotinib (alone or in combination with CDDP) to upregulate PD-L1 in normal tissues. We found that PD-L1 was transiently upregulated in the liver (but not in other tissues). These results have been added to the paper in supplementary Fig. 18e-h. We discussed the mechanism of toxicity of the combination regimens of ALK inhibitors and PD-(L)1 inhibitors that may be related to the temporary upregulation of PD-L1 in non-cancerous tissues.

Detailed comment No. 12 by reviewer #2: Typos: line 158- fibrosarcoma; line 168- others

Our response: Typos have been corrected.

Reviewers' Comments:

Reviewer #1:

Remarks to the Author:

The authors have fully responded to all queries previously raised. As such, the conclusion of the manuscript is fully supported by the large body of experimental evidence and it will be of high interest in the scientific community.

No further queries nor requests.

Reviewer #2:

Remarks to the Author:

The revised manuscript from Liu et al has superbly responded to all of my concerns in the original submission. They have convincingly demonstrated in multiple in vivo models of lung cancer that the combination of crizotinib with chemotherapy, such as platinum, induces an immune-mediated tumor cell killing that is not observed with either agent alone. The manuscript provides strong preclinical evidence for the efficacy of the combination of crizotinib/chemotherapy with immune checkpoint inhibition in the treatment of NSCLC. I think the authors have demonstrated the importance of these findings in understanding the mechanisms of immune-mediated killing and immune-mediated toxicity with ALK inhibitors. I believe this manuscript is a good fit for Nature Communications.

Point-by-point reply to reviewer #1

General remarks by the referee: The authors have fully responded to all queries previously raised. As such, the conclusion of the manuscript is fully supported by the large body of experimental evidence and it will be of high interest in the scientific community. No further queries nor requests.

Our response: We thank the reviewer for the positive comments. No further action is necessary on our side.

Point-by-point reply to reviewer #2

General remarks by the referee: The revised manuscript from Liu et al has superbly responded to all of my concerns in the original submission. They have convincingly demonstrated in multiple in vivo models of lung cancer that the combination of crizotinib with chemotherapy, such as platinum, induces an immune-mediated tumor cell killing that is not observed with either agent alone. The manuscript provides strong preclinical evidence for the efficacy of the combination of crizotinib/chemotherapy with immune checkpoint inhibition in the treatment of NSCLC. I think the authors have demonstrated the importance of these findings in understanding the mechanisms of immune-mediated killing and immune-mediated toxicity with ALK inhibitors. I believe this manuscript is a good fit for Nature Communications.

Our response: We thank the reviewer for his/her positive evaluation of our work. No further action is necessary on our side.